# Dietary Factors and Breast Cancer Prognosis among Breast Cancer Survivors: A Systematic Review and Meta-Analysis of Cohort Studies

**DOI:** 10.3390/cancers13215329

**Published:** 2021-10-23

**Authors:** Sin-Hye Park, Tung Hoang, Jeongseon Kim

**Affiliations:** 1Department of Food Science and Nutrition, Hallym University, Chuncheon 24252, Kangwon-do, Korea; shpark88@hallym.ac.kr; 2Department of Cancer Biomedical Science, National Cancer Center Graduate School of Cancer Science and Policy, Goyang 10408, Gyeonggi-do, Korea; hoangtunghup@gmail.com

**Keywords:** breast cancer, mortality, prognosis, dietary factors, breast cancer survivors

## Abstract

**Simple Summary:**

While most systematic reviews have focused on the association between dietary factors and breast cancer incidence, this current study focuses on the association between comprehensive dietary factors and breast cancer prognosis among breast cancer survivors by systematic review and meta-analysis. We reviewed a total of 63 cohort studies to assess the association between dietary factors and breast cancer prognosis by subgroup analysis with prediagnostic or postdiagnostic dietary intake, menopausal status, and dietary or supplementary micronutrient intake. We found that unhealthy dietary patterns, including the intake of beer and saturated fat, exacerbated the risk of breast cancer prognosis; however, the supplementation of most vitamins was desirable for breast cancer prognosis. Therefore, this study’s systematic review and meta-analysis provide useful dietary information for the development of dietary guidelines/recommendations to improve prognosis among breast cancer survivors.

**Abstract:**

Few studies have summarized the association between dietary factors and breast cancer (BC) prognosis among breast cancer survivors (BCS). Therefore, we carried out a systematic review and meta-analysis to determine the associations between dietary factors and BC prognosis among BCS. We performed a literature search in PubMed and Embase to investigate the association between dietary factors and BC prognosis. We applied a random-effects model to compute the hazard ratio/relative risk and their 95% confidence intervals and heterogeneity (Higgins I^2^) and to generate forest plots using STATA. Among the 2279 papers identified, 63 cohort studies were included in the systematic review and meta-analysis. Our main finding was that higher consumption of beer and saturated fat negatively affected BC prognosis. However, the intake of lignans, fiber, multivitamins, and antioxidants was negatively associated with the risk of mortality. Furthermore, we performed subgroup analyses by menopausal status and dietary or supplementary micronutrient intake. Most trends were similar to the main findings; in particular, the vitamin C, vitamin D, and vitamin E supplements decreased the risk of mortality. This study’s current systematic review and meta-analysis provide comprehensive dietary information for the development of dietary guidelines/recommendations to improve prognosis among BCS.

## 1. Introduction

Breast cancer (BC) is the most commonly diagnosed cancer among women worldwide and is the leading cause of cancer death, followed by colorectal cancer and lung cancer [1]. With advances in the early detection of BC cases and improved surgery and treatment, BC survival rates have been improved [2]. It is, therefore, important to investigate how lifestyle interventions, such as diet and physical activity, influence outcomes in cancer survivors [2]. The World Cancer Research Fund/American Institute for Cancer Research (WCRF/AICR) regularly releases reports on the extent of the relationship between diet, nutrition, and physical activities and BC survivors by reviewing a continuous update project (CUP) [3]. A recent CUP report suggested that higher consumption of foods containing fiber [4,5,6,7,8,9,10] and soy [11,12,13,14] and physical activity [15,16,17,18,19,20,21,22,23] may reduce the risk of all-cause mortality among breast cancer survivors (BCS). Moreover, BCS who consume a diet higher in total fat or saturated fatty acids may have a high risk of all-cause mortality [5,6,24,25]. However, due to a limited number of high-quality studies, these conclusions are not firm. Moreover, most previous findings have been analyzed with a focus on overall mortality rather than on either BC-specific mortality or recurrence. The report, furthermore, contained overall study types of epidemiological studies, such as case–control studies and randomized controlled trials (RCTs) as well as cohort studies, published up to 30 June 2012 [3]. Accordingly, the association between dietary factors and BC prognosis must be analyzed with up-to-date evidence from prospective cohort studies to provide stronger recommendations for BCS. As most observational studies have selection bias due to participant recruitment methods, nutritional epidemiological analyses generally prefer prospective cohort studies [26,27].

Several research teams have also recently published systematic reviews and meta-analyses on the associations between foods/nutrients and cancer prognosis. Some of these meta-analyses have investigated the association between dietary intake of fibers, soy foods, and isoflavones and BC survival in a dose-dependent manner [28,29,30]. Moreover, saturated fat intake rather than total fat intake has been found to be significantly associated with BC-specific mortality [31]. However, these previous studies included evidence from cohort and (nested) case–control studies. Moreover, the small effect sizes of the studies could not be resolved due to the limited number of studies, and relatively high heterogeneity was noted among the studies. Additionally, these studies did not assess publication bias, so we cannot make any firm conclusions to guide BC patients.

Therefore, we aim to comprehensively investigate the association between dietary factors and BC prognosis, including BC-specific mortality and recurrence as well as overall mortality, using a meta-analysis. Furthermore, we plan to assess the associations between dietary factors and BC prognosis with the more recent evidence from cohort studies rather than case–control studies. We also analyze whether the association between dietary factors and BC prognosis is changed by subgroup analysis, with three criteria: a timeframe of intake assessment (before vs. after BC diagnosis), menopausal status (premenopausal vs. postmenopausal), and the route of micronutrient intake (dietary vs. supplement use).

## 2. Materials and Methods

### 2.1. Literature Search and Selection Criteria

We searched and identified eligible cohort studies that investigated the association between dietary factors and BC prognosis on PubMed and Embase (from inception to 17 November 2020) by using the following keywords: “(diet or dietary or intake or consumption or nutrition or nutrients or macronutrients or micronutrients) and (breast cancer or breast neoplasms) and (survival or mortality or death or recurrence)”. Prior to searching the literature, all keywords were defined by medical subject headings (MeSH). An additional search was conducted based on a review of the references from eligible articles.

The inclusion criteria for the articles were as follows: (1) the design was a cohort study; (2) the population was breast cancer survivors; (3) the exposure of interest was any kind of dietary intake (single food items, nutrients, dietary patterns, and dietary scores/indexes); (4) the outcome of interest was all-cause mortality, breast cancer-specific mortality, or recurrence; (5) a full-text version of the article was available; and (6) the hazard ratio (HR) or relative risk (RR) and their 95% confidence intervals (CIs) were provided. The language was restricted to English. For duplicated or shared results from the same cohort study, the larger population or longer follow-up duration was included in the final meta-analysis. Two authors (S.-H. Park and T. Hoang) independently reviewed the literature search and study selection. Any disagreements were resolved by discussion with a third reviewer (J. Kim).

### 2.2. Data Extraction

Two investigators (S.-H. Park and T. Hoang) independently extracted the data according to the Preferred Reporting Items for Systematic Reviews and Meta-Analyses statement (PRISMA) [32]. Extracted data from full-text articles included the year of publication, name of the first author, project name, follow-up period, country, time point of dietary intake (before or after BC diagnosis), BC stage, age at baseline, sample size, route of consumption (intake of single food items, macronutrients, micronutrients, dietary patterns, or dietary scores/indexes), and outcome (all-cause mortality, BC-specific mortality, or recurrence; HR/RR with the corresponding 95% CI for each group of dietary exposure and covariates in each study).

### 2.3. Quality Assessment

The Newcastle-Ottawa Scale (NOS) for cohort studies was used to assess the quality of the studies included in the final analysis [33]. The NOS assessment system ranged from a score of 0 to 9 and included subscales for the selection of study groups, comparability of groups, and ascertainment of the outcome of interest. In the current study, all studies were considered to have scores of more than 6 for the final meta-analysis.

### 2.4. Data Analysis

We investigated the association between dietary factors (single food items, macronutrients, micronutrients, dietary patterns, and dietary scores/indexes) and BC prognosis (all-cause mortality, BC-specific mortality, or recurrence) by using adjusted HR or RR with 95% CIs for the main analysis. At least two studies on each dietary factor were available for the meta-analyses. We also conducted subgroup analyses based on prediagnostic or postdiagnostic dietary intake, menopausal status, and route of consumption (dietary or supplementary micronutrient intake). This current study was registered in the International Prospective Register of Systematic Reviews (PROSPERO; CRD42020222521).

### 2.5. Statistical Analysis

The main and subgroup analyses were performed by applying a random-effects model that combined the multivariable-adjusted HR/RR of the highest quartile compared with the lowest quartile. The main findings are shown in forest plots. We evaluated heterogeneity in between-study variation using Higgins I^2^. The Higgins I^2^ value was used to measure the percentage, and an I^2^ value greater than 50% suggested substantial heterogeneity [34]. We used STATA SE version 14.0 (StataCorp, College Station, TX, USA) for statistical analysis. Publication bias was not considered because there was an insufficient number of studies included to properly assess the funnel plot.

## 3. Results

### 3.1. Selection of Eligible Studies and Quality Assessment

Figure 1 shows the flow diagram for the identification of eligible studies according to PRISMA guidelines. Of the 2279 studies identified from PubMed and Embase, 2084 papers remained after the removal of duplicate records. Among them, 1979 papers were excluded by reviewing the title or abstract. In total, 105 papers were included in the assessment of eligibility. Among them, 65 papers were additionally excluded due to an inappropriate exposure and/or outcome (*n* = 22), inappropriate population (*n* = 4), no full-text (*n* = 5), a lack of HR or RR values (*n* = 4), systematic review and/or meta-analysis (*n* = 11), review article (*n* = 1), study protocol (*n* = 2), case–control study (*n* = 3), RCT (*n* = 6), clinical trial (*n* = 3), paper report (*n* = 1), clinical pilot study (*n* = 1), case report (*n* = 1), or overlap (*n* = 1). Moreover, 24 additional papers were identified by a manual search. Finally, 64 studies were included in the analysis [5,7,8,9,10,11,13,14,19,35,36,37,38,39,40,41,42,43,44,45,46,47,48,49,50,51,52,53,54,55,56,57,58,59,60,61,62,63,64,65,66,67,68,69,70,71,72,73,74,75,76,77,78,79,80,81,82,83,84,85,86,87,88]. Of these, one study was removed due to low study quality, according to the NOS. Thus, a total of 63 studies were ultimately included in the systematic review and meta-analysis.

### 3.2. Study Characteristics

The characteristics of the included studies are summarized in Appendix A. All studies were published between 1991 and 2020. The studies had been performed in the following countries: USA (*n* = 16), Canada (*n* = 4), Denmark (*n* = 3), China (*n* = 3), Germany (*n* = 2), Australia (*n* = 1), Sweden (*n* = 1), Europe United (*n* = 1), UK (*n* = 1), Italy (*n* = 1), Ireland (*n* = 1), and Japan (*n* = 1). Among a total of 120,167 BCS, 9659 premenopausal women and 35,366 postmenopausal women were included. The total number of deaths was 24,124, the number of BC-specific deaths was 28,030, and the number of recurrences was 4963. The mean/median follow-up period was 7.2 ± 2.72 years (standard deviation (SD)). The mean/median age of the population was 59.7 ± 6.25 years (SD) at baseline. Data on exposure were available from 32 studies for prediagnostic intake only [5,7,8,35,36,37,38,39,40,42,44,45,46,48,56,57,58,59,60,61,62,69,72,73,76,77,78,80,81,83], 21 studies for postdiagnostic intake only [9,10,11,13,14,41,49,50,51,52,53,54,55,68,70,75,82,85,86,87], and 13 studies for both prediagnostic and postdiagnostic dietary intake [8,41,52,63,64,65,67,70,74,79,84,88]. Regarding the subgroup analysis, data on exposure were available from 12 studies for premenopausal women [19,36,37,39,40,43,44,46,48,63,73,80] and 19 studies for postmenopausal women [5,19,36,38,39,40,43,44,46,48,59,60,69,70,72,73,80,84,87]. Data on outcomes were available from 7 studies for all-cause mortality only [5,8,43,54,72,79,85], 6 studies for BC-specific mortality only [7,35,36,39,64,82], 4 studies for recurrence only [40,46,77,81], 21 studies for both all-cause mortality and BC-specific mortality [10,19,41,44,45,50,51,56,57,58,60,61,63,67,70,73,74,75,76,78,80,86,87], 3 studies for both all-cause mortality and recurrence [11,42,53], 4 studies for both BC-specific mortality and recurrence [37,38,48,62], and 15 studies for all three factors of all-cause mortality, BC-specific mortality, and recurrence [9,13,14,47,49,52,55,59,65,66,68,69,71,84,88].

Appendix A shows the quality assessment of an individual study according to the NOS. Most studies achieved a score of at least 6 out of 9, which is considered good quality. However, one study [25] did not achieve a score for good quality (score = 5) and was excluded from the final analysis. All studies obtained dietary information via a food frequency questionnaire (FFQ).

### 3.3. Main Analysis

Table 1 displays the pooled estimates for the overall (prediagnostic and/or postdiagnostic) single food items (*n* = 32), macronutrients (*n* = 21), micronutrients (*n* = 46), dietary patterns (*n* = 2), and dietary indices (*n* = 13) and the association with BC prognosis. Figure 2 also shows forest plots for the main analysis with the association between dietary factors and BC prognosis. Using a random-effects model, the highest consumption of beer exacerbated the risk of recurrence by 42% (HR/RR = 1.42, 95% CIs = 1.03 to 1.95) [37,81]. Regarding nutrients, while the intake of saturated fat promoted the risk of BC-specific mortality with HR/RR = 1.82, 95% CIs = 1.26 to 2.62 [7,10,36,39], the intake of fiber [5,8,9,10] reduced the risk of all-cause mortality by 33% (0.67, 0.52 to 0.87). Moreover, we found that there were negative associations between the intake of multivitamins [52,53,55,68] and antioxidants [53,68], including niacin [8,45,54], vitamin D [8,10,54,68,72,85], and vitamin E [8,43,53,54,55,68], calcium [8,10,54,76], methionine [8,45], and isoflavone [11,13,14,44,60,79]^,^ and all-cause mortality. In particular, the intake of vitamin C, vitamin E [52,53,55,68], and isoflavones [11,14,40,48] were significantly associated with a lower risk of recurrence by 16%, 22%, and 21%, respectively. The intake of vitamin C [7,36,55,61,68], vitamin E [36,55,68], and lignans [44,73] were negatively associated with BC-specific mortality by 18%, 16%, and 16%, respectively. Based on the intake of these single foods/nutrients, we evaluated whether healthy dietary patterns and unhealthy dietary patterns contribute to prognosis. As a result, a healthy dietary pattern was negatively associated with all-cause mortality by 24% (HR/RR = 0.76, 95% CIs = 0.60 to 0.95), while an unhealthy dietary pattern was positively associated with all-cause mortality by 43% (1.43, 1.17 to 1.76) [41,47,69]. Unfortunately, healthy- and unhealthy dietary patterns were not shown to have a significant association with BC-specific mortality or recurrence. Furthermore, patients in the highest category of the CHFP [86] and HEI [50,51,70,86] had a 30% and 23% lower risk of all-cause mortality compared with patients in the lowest category; in particular, patients in the highest category of the CHFP also had a 35% lower risk of BC-specific mortality (HR/RR = 0.65, 95% CIs = 0.50 to 0.86) [86]. However, there was no significant association of the highest category of American Cancer Association (ACS) guideline adherence [74] and dietary inflammatory index (DII) [80,87] with the risk of mortality.

### 3.4. Subgroup Analyses

#### 3.4.1. Different Intake Time Points of Foods/Nutrients

We examined the association between dietary factors and BC prognosis according to prediagnostic dietary intake (24 single food items, 16 macronutrients, 32 micronutrients, 2 dietary patterns, and 2 dietary indexes) and postdiagnostic dietary intake (24 single food items, 11 macronutrients, 13 micronutrients, 2 dietary patterns, and 10 dietary indexes), which are presented in Table 1. Figure 3A,B shows the forest plots for the association between prediagnostic- and postdiagnostic dietary factors and BC prognosis. We found several patterns indicating a positive association between the prediagnostic intake of saturated fat and BC-specific mortality, with HR/RR = 2.03, 95% CIs = 1.26 to 3.27 [7,36,39]. In contrast, the prediagnostic intake of calcium [8,76] and folate [45,58] was associated with all-cause mortality, with HR/RR (95% CIs) of 0.71 (0.51 to 1.00) and 0.83 (0.68 to 1.00), respectively. Moreover, the prediagnostic intake of protein [7,39], vitamins [36,45,61], including thiamin and vitamin C, and lignans reduced the risk of BC-specific mortality by 47%, 52%, 29%, and 16%, respectively. Additionally, drinking beer contributed to promoting the risk of recurrence by 42% (HR/RR = 1.42, 95% CIs = 1.03 to 1.95) [37,81]. Based on these single food items/nutrients, a healthy dietary pattern was not associated with BC prognosis [47,69]; however, an unhealthy dietary pattern was significantly associated with all-cause mortality (HR/RR = 1.40, 95% CIs = 1.10 to 1.78) [41,47,69]. Regarding dietary factors by postdiagnostic intake, a higher intake of fiber was associated with a 36% lower risk of all-cause mortality [5,9,10]. Moreover, the postdiagnostic intake of multivitamins, particularly vitamin E [52,53,55,68], and isoflavones [11,14] was significantly associated with a 12%, 18%, and 21% lower risk of all-cause mortality and a 10%, 22%, and 25% lower risk of recurrence, respectively. Vitamin C intake was also associated with a 16% reduction in the risk of recurrence (HR/RR = 0.84, 95% CIs = 0.73 to 0.96) [53,55,68]. Furthermore, vitamin D intake was significantly associated with a 14% reduction in the risk of all-cause mortality (HR/RR = 0.86, 95% CIs = 0.78 to 0.95) [10,54,68,85]. Although useful information on foods/nutrients intake was available, we did not assess the association between dietary patterns and prognosis because only one study was included for each category of exposure and outcome. While we failed to assess dietary patterns, we found that patients in the highest category of the CHFP [86] had a 30% lower risk of all-cause mortality (HR/RR = 0.70, 95% CIs = 0.57 to 0.87) and 35% lower risk of BC-specific mortality (0.65, 0.50 to 0.86) than those in the lowest category. Moreover, the patients with a high HEI had an adverse association with all-cause mortality by 23% [50,51,86]. Therefore, we summarized that the prediagnostic consumption of beer and saturated fat would negatively affect BC prognosis. In contrast, prediagnostic intake of thiamin, calcium, folate, and lignans may protect against an unfavorable BC prognosis. Additionally, the postdiagnostic intake of protein, fiber, multivitamins, including vitamin C, vitamin D, and vitamin E, and isoflavones may improve BC prognosis. However, since unhealthy dietary patterns were mainly assessed with respect to the prediagnostic intake of foods/nutrients, we failed to assess the prognostic association with postdiagnostic unhealthy dietary patterns. In contrast, dietary scores/indexes were more likely to be assessed with respect to postdiagnostic intake rather than prediagnostic intake.

#### 3.4.2. Menopausal Status

The subgroup analysis for the associations between dietary factors and prognosis by menopausal status is presented in Table 2. Figure 3C shows the forest plots of the main findings stratified by premenopausal and/or postmenopausal status. In general, most of the population were postmenopausal, accounting for approximately 79% of the whole population, whereas premenopausal women accounted for 1/5 of the total population. Thus, most characteristics were mainly assessed with postmenopausal women. The intake of rye bread and cheese was positively associated with all-cause mortality among postmenopausal women, with a 13% and 16% increased risk observed, respectively [84]. In contrast, the intake of oatmeal/muesli reduced the risk of all-cause mortality by approximately 22% among postmenopausal women [84]. Moreover, the intake of lutein/zeaxanthin substantially lowered the risk of all-cause mortality for postmenopausal BCS, with HR/RR = 0.59, 95% CIs = 0.41 to 0.84 [5,43]. Interestingly, the intake of lignans, which are known sources of phytoestrogens, led to different outcomes in premenopausal and postmenopausal women; there was an increased risk of mortality observed in premenopausal women but a decreased risk observed in postmenopausal women, with HR/RRs (95% CIs) of 1.26 (1.06 to 1.50) and 0.83 (0.72 to 0.96), respectively [44,73]. Accordingly, even though postmenopausal women comprised the majority in most studies, the results indicate that the intake of oatmeal/muesli, lutein/zeaxanthin, and lignans showed a negative association with BC prognosis. Further study is needed to explain why lignan intake affects prognosis differently in premenopausal and postmenopausal BCS.

#### 3.4.3. Dietary or Supplementary Micronutrient Intake

Finally, we performed a subgroup analysis according to micronutrient intake from daily diet or supplements to determine whether the intake of nutrients by different routes contributes to BC prognosis. We have summarized the association between dietary or supplementary nutrient intake and prognosis in Table 3, and we also show the forest plots in Figure 3D,E. Since most studies did not specify in detail whether nutrients were consumed from diet or supplements, we had to perform this subgroup analysis with little evidence. Overall, multivitamins and antioxidants were entirely consumed from supplements, and those who took multivitamin supplements had 12% and 10% lower risks of all-cause mortality and recurrence, respectively [52,53,55,68]. Among the vitamin and antioxidant groups, BCS were more likely to consume niacin, vitamin C, and folate from their diet; furthermore, vitamin C, vitamin D, and vitamin E were likely to be consumed from supplements. Regarding the intake of minerals, most BCS seemed to consume calcium, magnesium, methionine, and isoflavones from their diet. Dietary intake of niacin [8,45] and folate [5,45,58] reduced the risk of all-cause mortality by 26% and 34%, respectively. Likewise, patients consuming dietary calcium [8,10,76], magnesium [8,76], and methionine [8,45] also had 28%, 34%, and 31% lower risks of all-cause mortality, respectively. Additionally, dietary intake of isoflavones was adversely associated with recurrence by 22% (HR/RR = 0.78, 95% CIs = 0.65 to 0.93) [13,14,60]. As mentioned above, most BCS were prone to consuming vitamins via supplements, and the intake of major vitamins, including vitamin C, vitamin D, and vitamin E, were shown to improve prognosis. In particular, vitamin E supplementation led to substantially lower risks of poor prognostic outcomes, with HR/RRs (95% CIs) of 0.82 (0.73 to 0.92) for all-cause mortality [8,53,54,55,68], 0.84 (0.70 to 1.00) for BC-specific mortality [36,55,68], and 0.78 (0.64 to 0.95) for recurrence [53,55,68]. Since vitamin C is easily taken up from the diet, dietary vitamin C intake led to a 24% lower risk of BC-specific mortality (HR/RR = 0.76, 95% CIs = 0.59 to 0.97) [7,61] relative to supplement use (0.79, 0.61 to 1.02) [36,55,68]. However, vitamin C supplementation showed preventive effects on the risk of recurrence, with HR/RR = 0.84, 95% CIs = 0.73 to 0.96 [53,55,68]. Taken together, the findings indicate that the intake of vitamin C, vitamin D, and vitamin E contributed to a favorable prognosis among BCS. Moreover, the intake of dietary calcium, magnesium, and isoflavones exerts substantial preventive effects against poor prognosis.

## 4. Discussion

In this systematic review and meta-analysis of 63 cohort studies, we aim to investigate the association between dietary factors and BC prognosis among BCS. We found various associations between dietary factors and all-cause mortality or BC-specific mortality or recurrence. We briefly summarize the main findings as follows: (1) A prediagnostic beer consumption exacerbated the risk of all-cause mortality and recurrence. (2) A higher intake of prediagnostic saturated fat increased the risk of BC-specific mortality. (3) The intake of prediagnostic calcium, folate, and lignans was associated with a reduced risk of mortality. (4) The postdiagnostic intake of fiber, antioxidants, and multivitamins, including vitamin C, vitamin D, and vitamin E, was negatively associated with BC prognosis. (5) The postdiagnostic intake of vitamin C, vitamin E, and isoflavones was negatively associated with recurrence. (6) A healthy/prudent dietary pattern was negatively associated with all-cause mortality; however, an unhealthy/Western dietary pattern was positively associated with all-cause mortality. (7) The highest category of the CHFP and HEI was significantly associated with all-cause mortality. (8) After subgroup analysis, the dietary intake of lignans increased the risk of mortality among premenopausal women but seemed to decrease the risk of mortality among postmenopausal women. (9) Most vitamins and antioxidants were consumed by supplementation rather than in the diet, which contributed to a better prognosis for BCS; thus, BCS with a dietary pattern enriched in vegetables, fruits, and soy products may achieve long-term survival.

It is known that the CUP report is the most important study among investigations of the association between modifiable lifestyle factors and BC prognosis [3]. Accordingly, we have primarily compared the main findings between our current study and the CUP report to determine whether there are any differences. Their report suggests that the consumption of foods containing fiber and soy isoflavones reduces the risk of overall mortality, while the consumption of foods containing high (saturated) fat may increase the risk of all-cause mortality [3]. However, this report has some limitations. For example, they assessed overall epidemiological study designs, including case–control and RCTs as well as cohort studies, so firm conclusions for BCS could not be drawn. Additionally, their results mainly focused on BC survival, not recurrence, for which evidence was published until 2012. The most important aspect of the current study is that we have overcome these limitations, assessed overall prognostic outcomes, and included more recently published cohort studies to provide the CUP report.

Regarding the association between the intake of fiber and soy foods and BC prognosis, our results showed a similar trend without heterogeneity. Additionally, these dietary factors have been evaluated by some previous meta-analyses [28,29,89,90], which showed concordance with the current results. Moreover, our results suggest that postdiagnostic isoflavone intake is associated with a 21% lower risk of recurrence and an 11% lower risk of BC-specific mortality with more recent cohort studies, which were not indicated in the CUP report. However, the correlation between isoflavone intake and prognosis showed moderate heterogeneity, with an I^2^ of 50.1% for recurrence and 44.1% for all-cause mortality. Since our findings had a relatively higher heterogeneity, we estimate there were differences in hormonal receptor status and ethnicity. There is some evidence that estrogen receptor (ER)/progesterone receptor (PR) negativity shows an inverse association with mortality and recurrence in postmenopausal women who are not treated with hormonal therapy [13,14,60,79]. There is a potential explanation for the beneficial effect of isoflavones, mostly in women with ER/PR-negative BC: isoflavones are more likely to bind to the ER and act as estrogen agonists when circulating estrogen levels are low [60]. Moreover, there are differences in isoflavone intake between Asian and Western women; for example, American women consume an average of 3.2 mg isoflavones/day, while Chinese women consume an average of 45.9 mg isoflavones/day [14]. In addition, there are possible reasons that isoflavones have shown a modest effect on BCS, namely, conflicting in vivo and in vitro data, accounting for the stimulation of cell proliferation [91,92,93,94]. There were some findings reported in previous meta-analyses that are consistent with our findings [29,30]. Although these prior studies considered overall epidemiological study designs, they suggested that there was some publication bias [29] Moreover, these studies performed dose–response meta-analyses, and the findings indicated that a 10 mg/day increase in the intake of soy isoflavones was associated with a 7–9% decrease in the risk of mortality, and a 5 g/day increase in soy protein was associated with a 12% decrease in the risk of BC-specific mortality [30]. Therefore, further study is needed to examine the association between isoflavone intake and prognosis according to hormonal receptor status, hormone therapy administered, and ethnicity.

Finally, the CUP report indicated that the association between a higher intake of total/saturated fat and mortality risk was unclear. To date, even though several meta-analyses have tried to investigate this inconsistency, they have mainly focused on BC incidence rather than prognosis [95,96]. We found one meta-analysis that was very similar to ours, and the results indicated that the intake of saturated fat significantly increased the risk of BC-specific mortality, with relatively low heterogeneity (I^2^ = 15%) [31]. Furthermore, they examined publication bias (Begg’s test, *p*-value = 0.156), while we did not assess publication bias due to insufficient study numbers, which is a limitation. There are some potential biological mechanisms by which fat intake stimulates malignant mammary cell growth by increasing circulating estrogens [97,98], and saturated fat intake enhances tumor growth by increasing low-density lipoprotein and cholesterol levels and boosting the inflammatory response [31,99,100]. Therefore, we determined that saturated fat is a critical risk factor for BC prognosis.

Herein, we have summarized not only the findings of CUP reports but also various dietary factors that influence the BC prognosis. First, we have shown that prediagnostic beer consumption increases the risk of mortality and recurrence [37,67,81]. Some meta-analyses have investigated the association between alcohol subtypes and BC risk; however, they all focused on the incidence [101,102]. There was also one case–control study that examined the association between the prediagnostic consumption of various alcoholic beverages and mortality [103]. Therefore, the strength of our meta-analysis is that this is the first study to summarize the relationship between alcohol consumption subtypes and BC prognosis with recent cohort studies; however, the limitation is the small effect size. Although no mechanistic studies have elucidated that beer consumption directly affects prognosis in vitro and in vivo, we assume there are possible mechanisms by which alcohol consumption aggravates prognosis by promoting breast cancer cell proliferation, cell transformation, and tumor development via Brf1 gene expression and by increasing the levels of estrogen and androgen, making individuals susceptible to the risk of breast carcinogenesis [104,105,106].

Second, our findings suggest that the intake of prediagnostic calcium, folate, and lignans is adversely associated with mortality. Subgroup analysis revealed that calcium and folate are mostly consumed from diet rather than from supplementation, while the effects of lignan intake differed remarkably based on menopausal status, regardless of whether intake was from diet or supplementation. The intake of dietary calcium showed a trend similar to that reported in previous meta-analyses; however, these studies focused on BC incidence [107,108]. Additionally, calcium may reduce the risk of BC development by decreasing cell proliferation and differentiation [8,10,90,109,110]. This is the first study to examine the association between dietary calcium intake and the risk of mortality. Additionally, regarding the intake of dietary folate, our result was similar to that of a meta-analysis that examined the association between folate intake and prognosis with observational studies [111]. The studies exploring the dietary intake of folate included in the analysis had relatively high heterogeneity (I^2^ = 65.5%), so the possible underlying mechanisms remain unclear. However, we may assume that an inadequate level of circulating folate may foster tumor growth by activating oncogenes via DNA methylation [112]. In contrast, excessive folate may influence cancer development via epigenetic changes in oncogene-regulatory mechanisms [113]. Therefore, to provide nutritional guidance, we need to further analyze the association between folate intake and prognosis according to intake doses via a dose–response meta-analysis. Otherwise, we suggest that the intake of lignans have a protective effect in postmenopausal women than in premenopausal women and that our findings are in line with some previous meta-analyses and case–control studies [114,115,116]. However, these prior studies focused on incidence, not prognostic outcomes. Accordingly, our study has an advantage in that it is the first to investigate the association between the intake of lignans and prognosis by menopausal status with recent cohort studies. Lignan is a phytoestrogen that is structurally similar to estradiol [114]. We speculated that the more beneficial effect of lignans observed in postmenopausal women is because lignans may physiologically be more protective under low estradiol conditions. Moreover, there is evidence that lignans stimulate intestinal microflora and enterohepatic or circulating estrogen levels [4,117,118]. Regarding the evaluation of lignan intake, several previous studies detected blood enterolactone levels, a lignan metabolite [4,119]. Therefore, further analysis of enterolactone levels with subgroup analysis based on hormonal receptor status could be valuable for determining the biological effects of lignans on prognosis.

Third, we suggest that a higher intake of dietary fiber decreases the risk of mortality, similar to the CUP findings mentioned above. There are some potent mechanisms by which a higher intake of fiber can lead to lower serum estradiol levels, eliciting beneficial effects against BC [120]. Additionally, there is evidence to show that the protective effect of fiber varies according to the subtype, such as soluble vs. insoluble fiber. In particular, vegetable fiber has the strongest protective effect on mammary tumorigenesis due to the combination of insoluble (cellulose) and soluble (pectin) fiber [121,122]. The intake of dietary soluble fiber was also negatively associated with BC, which has been verified with epidemiological evidence [123,124]. In addition to the effect of dietary fiber, absorbed fiber can ferment in the gastric intestinal tract and have more effective bioavailability due to the anti-inflammatory action of butyrate and propionate [125,126,127]. Therefore, fiber-rich foods, in particular those that contain a 1:1 mix of both soluble and insoluble fibers and their fermented metabolites, may be a beneficial factor for BCS. Accordingly, additional studies are needed to examine whether different types of fiber may be attributed to prognosis to determine which fermented fiber metabolites are generated and to explore how much of an effect these metabolites exert on long-term survival.

Importantly, our current study suggests that the postdiagnostic intake of multivitamins and antioxidants, including vitamin C, vitamin D, and vitamin E, is beneficial for BC prognosis; in particular, vitamin C, vitamin E, and isoflavones was found to significantly reduce the risk of recurrence. However, there was relatively high heterogeneity among the studies for vitamin C (I^2^ = 58.9% for all-cause mortality and 28.5% for recurrence) and vitamin E (I^2^ = 51.4% for recurrence). There are two possible explanations: one is that supplementation with vitamin C and vitamin E during chemotherapy or radiotherapy treatment reduces the cytotoxic effect of tumor cells produced by chemotherapy agents; the other is that antioxidant supplementation protects against the damage from oxidative stress in tumor cells [128,129,130]. Given these reasons, we suggest caution when using antioxidant supplements during chemotherapy treatment for BC patients. We may have better guidance for BCS once we further assess the association between antioxidant supplementation and prognosis according to whether BCS were undergoing chemotherapy or finishing chemotherapy. Additionally, vitamin C from diet as well as supplementation had preventive effects against BC-specific mortality. Even though the studies included had very small numbers, the findings of one cohort study were consistent with our results [61]. The results suggested that vegetables, as a source of vitamin C, are beneficial for long-term survival; however, the association between vegetable intake and BC prognosis was not statistically significant [5,10,43]. Therefore, we need to further explore the types of food sources for vitamin C, which exerts a beneficial effect against BC-specific mortality.

Finally, we determined whether dietary patterns and dietary scores decrease the risk of mortality. Most dietary intake information was assessed from the intake before diagnosis for dietary patterns and from the intake after diagnosis for dietary scores/indexes. We divided the dietary patterns into healthy dietary patterns (prudent diets; rich with fruits, vegetables, and fibers) and unhealthy dietary patterns (Western diets; rich with meats, sweets, desserts, and high-fat dairy products) to determine the association. The results indicated that a healthy dietary pattern was negatively associated with all-cause mortality by 24%, and an unhealthy dietary pattern was positively associated with all-cause mortality by 43%. Lifestyle patterns were determined according to each dietary pattern: women with higher healthy diet scores were more likely to participate in physical activities, have higher educational levels, and use vitamin supplements and were less likely to smoke; however, women with higher Western diet scores were more likely to be obese and smoke and less likely to use vitamin supplements [41,47,69]. Given that differences in lifestyle patterns were found according to the different dietary patterns, we consider these findings to coincide. This current study included recently published cohort studies; however, the study population was limited and mainly comprised Americans and Europeans. Therefore, the relationship between dietary patterns and prognosis warrants further investigation, particularly in Asian populations, to update evidence. Moreover, regarding dietary scores/indexes, we found that the CHFP and HEI were related to mortality risk among BCS. Most studies evaluated dietary quality in terms of cancer prevention with the HEI or DASH, which were developed for Americans; however, these are not specific to cancer and show beneficial effects on CVD and diabetes in the Chinese population [131]. Accordingly, the Chinese Nutrition Society and Ministry of Health developed CHFP guidelines for 11 food groups of general food items consumed by Chinese [132], and it was shown that women with higher adherence to the CHFP tended to have lower mortality risk [86]. Overall, these dietary quality scores/indexes are limited to specific ethnicities; therefore, additional studies are needed to establish high-quality, universal dietary tools to guide long-term survival in BCS.

To our knowledge, this study is the most comprehensive systematic review and meta-analysis of cohort studies investigating the association between dietary factors and BC prognosis among BCS. Most studies have focused on BC incidence [30,133,134], so the strength of the current study lies in its focus on overall BC prognosis. Although the CUP report provided updated risk factors for BC prognosis [3], our current study contained more recent cohort studies on the association between dietary factors and overall prognostic outcomes. Additionally, since we have performed subgroup analyses with various factors, such as prediagnostic- and postdiagnostic intake, menopausal status, and dietary- and supplementary micronutrient intake, we can suggest that BCS who are more likely to consume fruits, vegetables, fiber, and supplement use of multivitamins and antioxidants after diagnosis will have a better prognosis. Moreover, individuals at risk should be cautioned that they should not consume a large amount of saturated fats. However, this study has some limitations. Fewer than ten studies were included for each category of exposure or outcome, so we could not assess whether there was publication bias. Given that the study numbers were not sufficient to be generalized, we cannot provide strong dietary recommendations for BCS. Moreover, most dietary factors were not assessed with the same units between studies; for example, information was not synchronized relative to a dose or unit for intake frequency. Therefore, we need to conduct a dose–response meta-analysis with the suggested dietary factors from the current study.

## 5. Conclusions

In conclusion, we provide comprehensive evidence to support dietary recommendations that could improve prognosis among BCS. Based on our systematic review and meta-analysis, we recommend BCS have a higher intake of soy isoflavones and lignans, multivitamins, and antioxidant supplements and a lower intake of saturated fats to improve prognosis. The results of our systematic review and meta-analysis may provide useful dietary information for the development of dietary guidelines/recommendations to improve prognosis among BCS.

## Figures and Tables

**Figure 1 cancers-13-05329-f001:**
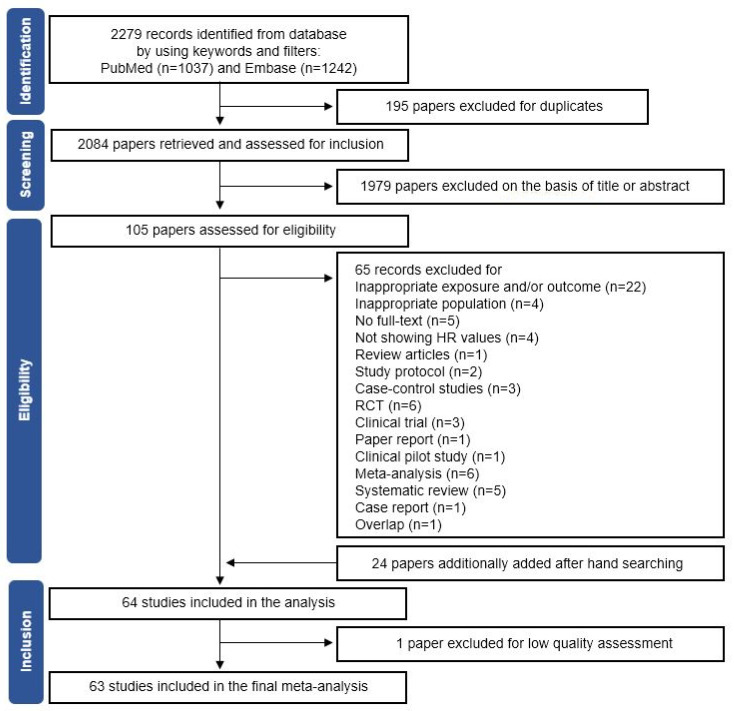
Flow chart of study selection. The flow chart shows the process used to select cohort studies for the systematic review and meta-analysis of the association between dietary factors and breast cancer prognosis among breast cancer survivors.

**Figure 2 cancers-13-05329-f002:**
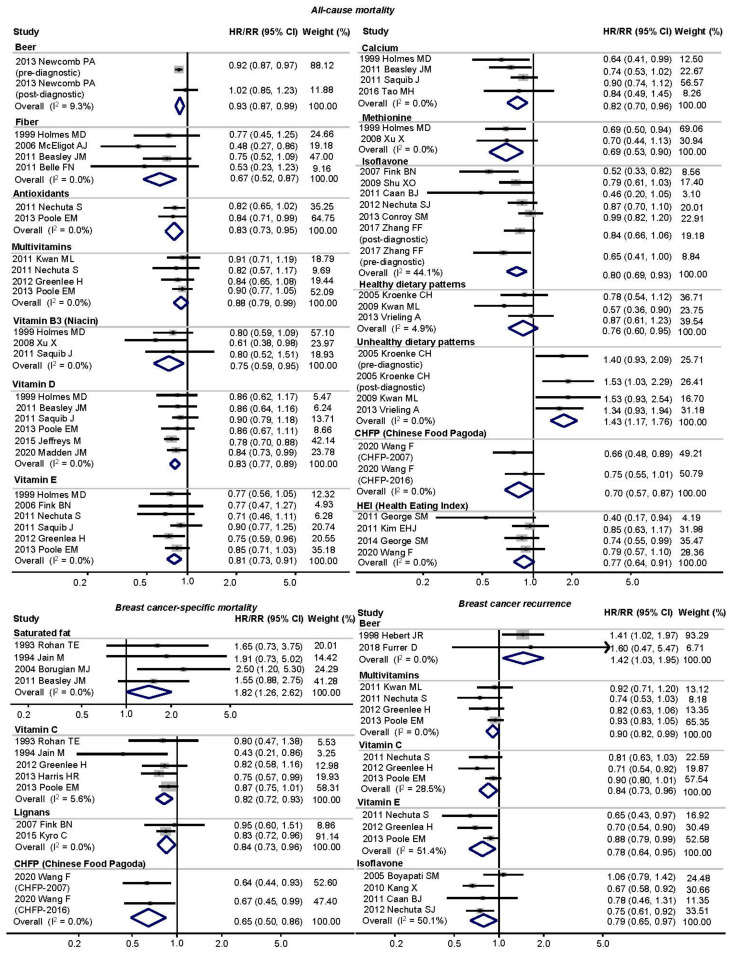
Forest plots for the roles of overall dietary factors and all-cause mortality among breast cancer survivors. Abbreviations: HR, hazard ratio; RR, relative risk; CI, confidence intervals.

**Figure 3 cancers-13-05329-f003:**
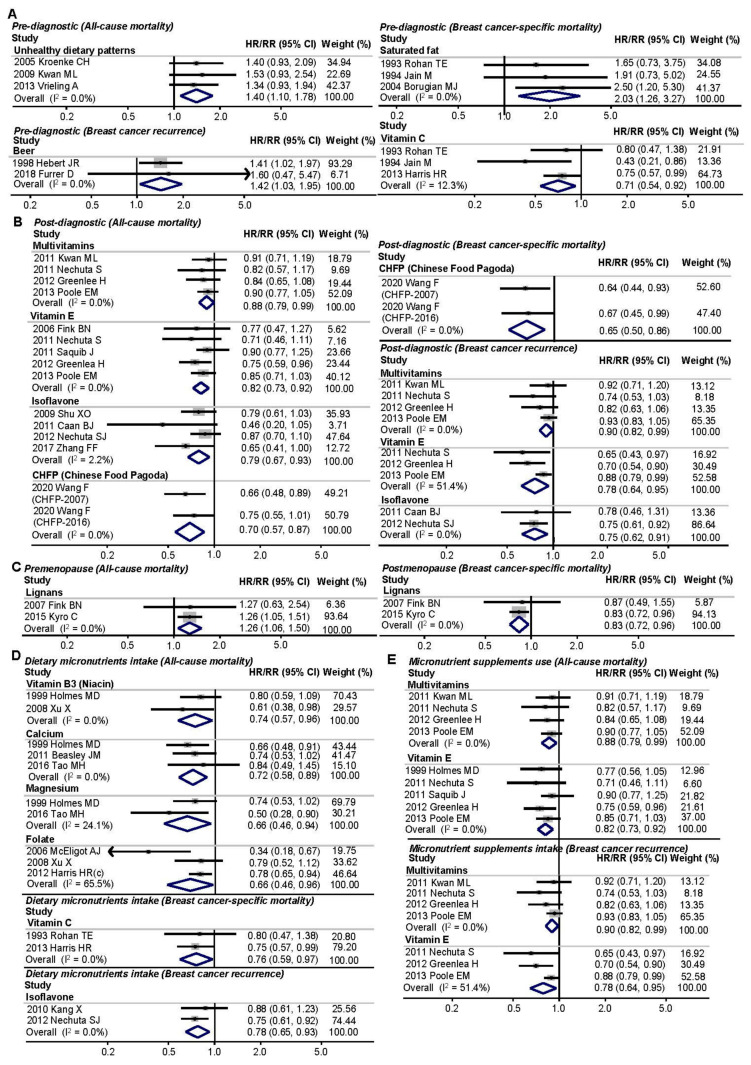
Forest plots of subgroup analysis for the associations between dietary factors and BC prognosis by prediagnostic (**A**) and postdiagnostic intake (**B**), menopausal status (**C**), and dietary (**D**) and/or supplement use of micronutrient intake (**E**) among breast cancer survivors. Abbreviations: HR, hazard ratio; RR, relative risk; CI, confidence intervals.

**Table 1 cancers-13-05329-t001:** Systematic review and meta-analysis of associations between diagnostic dietary factors and breast cancer prognosis among breast cancer survivors.

	All-Cause Mortality		Breast-Cancer-Specific Mortality		Breast Cancer Recurrence	
Overall	N (I^2^)	RR/HR (95% CI)	N (I^2^)	RR/HR (95% CI)	N (I^2^)	RR/HR (95% CI)
Single food items						
Alcohol	10 (46.6%)	0.92 (0.83 to 1.02)	13 (55.9%)	1.12 (0.96 to 1.31)	4 (26.2%)	1.20 (0.99 to 1.44)
Beer	2 (9.3%)	0.93 (0.87 to 0.99)	3 (67.5%)	1.14 (0.84 to 1.56)	2 (0.0%)	1.42 (1.03 to 1.95)
Wine	3 (0.0%)	0.88 (0.84 to 0.93)	3 (5.9%)	1.00 (0.90 to 1.10)	2 (84.5%)	0.80 (0.26 to 2.45)
Spirits	2 (0.0%)	0.90 (0.86 to 0.95)	2 (35.8%)	0.88 (0.74 to 1.05)		
Coffee	1 (NA)	1.12 (0.84 to 1.51)	1 (NA)	1.14 (0.71 to 1.83)		
Tea	2 (38.4%)	0.82 (0.54 to 1.26)	1 (NA)	1.02 (0.67 to 1.55)		
				1 (NA)	0.60 (0.29 to 1.27)	
Whole grain products	5 (0.0%)	1.01 (0.96 to 1.06)	4 (0.0%)	1.04 (0.98 to 1.10)	4 (14.1%)	1.01 (0.94 to 1.08)
Grains	2 (0.0%)	1.08 (0.94 to 1.25)	2 (0.0%)	1.13 (0.88 to 1.44)		
Oatmeal/muesli	2 (0.0%)	0.78 (0.63 to 0.97)	2 (0.0%)	0.92 (0.72 to 1.18)	2 (0.0%)	0.91 (0.71 to 1.16)
Rye bread	2 (0.0%)	1.13 (1.02 to 1.25)	2 (43.0%)	1.19 (1.00 to 1.42)	2 (38.4%)	1.11 (0.92 to 1.34)
Any fruits, fruit juices, and vegetables	1 (NA)	0.68 (0.42 to 1.09)				
Fruits and vegetables	3 (72.8%)	0.88 (0.60 to 1.30)	3 (0.7%)	1.09 (0.81 to 1.47)	1 (NA)	0.58 (0.25 to 1.36)
Fruits	3 (62.0%)	1.00 (0.67 to 1.48)	1 (NA)	1.39 (0.64 to 2.99)		
Fruits and fruit juices	1 (NA)	0.87 (0.57 to 1.35)				
Citrus fruits	1 (NA)	0.93 (0.61 to 1.42)				
Vegetables	4 (60.4%)	0.98 (0.68 to 1.40)	1 (NA)	0.96 (0.38 to 2.45)		
Cruciferous vegetables	2 (0.0%)	1.03 (0.83 to 1.28)	1 (NA)	0.95 (0.59 to 1.54)		
Leafy vegetables	1 (NA)	0.72 (0.41 to 1.24)				
Yellow vegetables	1 (NA)	0.90 (0.58 to 1.40)				
Dairy products	5 (40.6%)	1.04 (0.95 to 1.13)	4 (0.0%)	0.99 (0.93 to 1.05)	3 (0.0%)	0.98 (0.91 to 1.04)
High-fat dairy products					1 (NA)	1.09 (0.88 to 1.35)
Low-fat dairy products					1 (NA)	0.84 (0.69 to 1.04)
Butter/margarine/lard			1 (NA)	1.16 (0.86 to 1.58)	1 (NA)	1.30 (1.03 to 1.64)
Cheese	2 (0.0%)	1.16 (1.06 to 1.27)	2 (0.0%)	1.11 (0.92 to 1.35)	2 (0.0%)	1.18 (0.98 to 1.43)
Milk	2 (0.0%)	1.02 (0.96 to 1.08)	2 (0.0%)	1.00 (0.93 to 1.07)	2 (0.0%)	0.95 (0.88 to 1.02)
Yogurt	2 (0.0%)	0.91 (0.79 to 1.05)	2 (0.0%)	0.85 (0.71 to 1.03)	2 (0.0%)	0.99 (0.83 to 1.18)
Soy products	1 (NA)	1.03 (0.81 to 1.33)	1 (NA)	1.03 (0.71 to 1.50)		
Fish	1 (NA)	0.94 (0.62 to 1.43)			1 (NA)	0.93 (0.76 to 1.15)
Poultry	1 (NA)	0.60 (0.39 to 0.92)			1 (NA)	0.85 (0.69 to 1.05)
Red and processed meat	4 (66.8%)	0.89 (0.71 to 1.13)	4 (0.0%)	1.05 (0.83 to 1.31)	2 (0.0%)	1.04 (0.85 to 1.28)
Natural products	1 (NA)	0.95 (0.67 to 1.35)	1 (NA)	1.15 (0.69 to 1.94)		
Ginseng	1 (NA)	0.71 (0.52 to 0.98)			1 (NA)	0.70 (0.53 to 0.93)
Macronutrients						
Carbohydrates	2 (0.0%)	0.97 (0.73 to 1.29)	4 (0.0%)	1.01 (0.72 to 1.43)	1 (NA)	0.77 (0.27 to 2.19)
E-Carb			1 (NA)	1.70 (0.70 to 3.80)		
Fat	3 (83.1%)	1.52 (0.85 to 2.74)	4 (0.0%)	1.02 (0.83 to 1.25)		
Trans fat	1 (NA)	1.78 (1.35 to 2.32)	1 (NA)	1.42 (0.80 to 2.52)		
Saturated fat	2 (89.3%)	2.40 (0.78 to 7.38)	4 (0.0%)	1.82 (1.26 to 2.62)		
Saturated fat/total fat			1 (NA)	1.93 (1.00 to 3.74)		
Monounsaturated fat	1 (NA)	1.14 (0.86 to 1.52)	2 (0.0%)	1.01 (0.62 to 1.65)		
Polyunsaturated fat	1 (NA)	0.91 (0.70 to 1.19)	2 (0.0%)	1.19 (0.66 to 2.12)		
Omega-3 fatty acids	1 (NA)	1.00 (0.62 to 1.60)				
Linoleic fatty acids	1 (NA)	2.39 (1.21 to 4.69)				
Oleic fatty acids	1 (NA)	3.56 (1.67 to 7.59)				
18:2 trans fatty acid	1 (NA)	1.58 (1.03 to 2.43)				
Protein	2 (38.5%)	0.86 (0.62 to 1.19)	3 (63.8%)	0.72 (0.37 to 1.39)	1 (NA)	0.84 (0.69 to 1.03)
Soy protein	1 (NA)	0.71 (0.54 to 0.92)			1 (NA)	0.99 (0.73 to 1.33)
				1 (NA)	0.68 (0.54 to 0.87)	
Animal protein					1 (NA)	0.74 (0.61 to 0.91)
Vegetable protein					1 (NA)	1.29 (1.05 to 1.59)
Essential amino acids					1 (NA)	0.86 (0.71 to 1.05)
Branched-chain amino acids					1 (NA)	0.82 (0.68 to 1.00)
Pantothenic acid	1 (NA)	0.92 (0.67 to 1.29)				
Tryptophan	1 (NA)	0.63 (0.46 to 0.87)				
Fiber	4 (0.0%)	0.67 (0.52 to 0.87)	4 (0.0%)	0.75 (0.54 to 1.04)	1 (NA)	0.68 (0.27 to 1.70)
Micronutrients						
Multivitamins	4 (0.0%)	0.88 (0.79 to 0.99)	3 (0.0%)	0.89 (0.79 to 1.02)	4 (0.0%)	0.90 (0.82 to 0.99)
Antioxidants	2 (0.0%)	0.83 (0.73 to 0.95)	1 (NA)	0.86 (0.73 to 1.02)	2 (57.3%)	0.87 (0.73 to 1.04)
				1 (NA)	0.54 (0.27 to 1.04)	
Vitamin A	3 (36.4%)	0.97 (0.80 to 1.18)	3 (35.0%)	0.91 (0.64 to 1.30)	1 (NA)	1.07 (0.72 to 1.60)
Retinol	2 (36.9%)	0.96 (0.76 to 1.21)	1 (NA)	1.05 (0.56 to 1.95)		
Vitamin B	1 (NA)	0.87 (0.74 to 1.02)	1 (NA)	0.90 (0.74 to 1.10)	1 (NA)	0.88 (0.75 to 1.03)
Thiamin	2 (58.0%)	0.68 (0.45 to 1.02)	2 (0.0%)	0.48 (0.30 to 0.78)		
Riboflavin	2 (0.0%)	0.85 (0.65 to 1.11)	1 (NA)	0.72 (0.41 to 1.29)		
Niacin	3 (0.0%)	0.75 (0.59 to 0.95)	1 (NA)	0.61 (0.34 to 1.09)		
Vitamin B6	3 (0.0%)	0.91 (0.74 to 1.11)	1 (NA)	0.77 (0.44 to 1.36)		
Vitamin B12	1 (NA)	1.20 (0.80 to 1.81)	1 (NA)	1.10 (0.65 to 1.85)		
Vitamin C	8 (57.1%)	0.88 (0.76 to 1.01)	5 (5.6%)	0.82 (0.72 to 0.93)	3 (28.5%)	0.84 (0.73 to 0.96)
				1 (NA)	0.64 (0.32 to 1.27)	
Vitamin D	6 (0.0%)	0.83 (0.77 to 0.89)	3 (0.0%)	0.85 (0.72 to 1.01)	1 (NA)	0.82 (0.60 to 1.12)
Vitamin E	6 (0.0%)	0.81 (0.73 to 0.91)	3 (0.0%)	0.84 (0.70 to 1.00)	3 (51.4%)	0.78 (0.64 to 0.95)
				1 (NA)	0.55 (0.28 to 1.08)	
Carotenoids	1 (NA)	1.63 (1.06 to 2.50)	1 (NA)	1.93 (1.14 to 3.28)	1 (NA)	1.23 (0.76 to 1.96)
Carotenes	1 (NA)	0.96 (0.70 to 1.31)				
Alpha-Carotene	4 (0.0%)	1.01 (0.85 to 1.21)	1 (NA)	0.98 (0.59 to 1.64)		
Beta-Carotene	5 (51.7%)	0.96 (0.74 to 1.24)	4 (33.2%)	0.92 (0.63 to 1.36)	1 (NA)	0.89 (0.50 to 1.60)
Provitamin A	1 (NA)	0.58 (0.34 to 0.99)				
Beta-Cryptoxanthin	4 (61.2%)	0.88 (0.64 to 1.20)	1 (NA)	0.81 (0.45 to 1.45)		
Lutein/zeaxanthin	4 (54.4%)	0.77 (0.56 to 1.07)	1 (NA)	1.16 (0.62 to 2.19)		
Lycopene	5 (0.0%)	0.91 (0.76 to 1.09)	2 (0.0%)	1.51 (0.90 to 2.55)	1 (NA)	1.17 (0.35 to 3.89)
Selenium	3 (0.0%)	0.88 (0.68 to 1.14)	1 (NA)	0.90 (0.45 to 1.79)	1 (NA)	0.89 (0.53 to 1.49)
Betaine	1 (NA)	0.81 (0.54 to 1.20)	1 (NA)	0.72 (0.44 to 1.17)		
Caffeine	1 (NA)	0.77 (0.55 to 1.07)				
Calcium	4 (0.0%)	0.82 (0.70 to 0.96)	3 (31.4%)	0.77 (0.48 to 1.24)		
Magnesium	3 (50.1%)	0.76 (0.53 to 1.07)	1 (NA)	0.70 (0.33 to 1.49)		
Iron	2 (70.7%)	1.12 (0.61 to 2.03)				
Iodine	1 (NA)	0.90 (0.67 to 1.21)				
Potassium	1 (NA)	0.98 (0.69 to 1.38)				
Sodium	1 (NA)	0.79 (0.57 to 1.09)				
Zinc	3 (24.8%)	0.92 (0.73 to 1.15)	1 (NA)	0.82 (0.44 to 1.53)	1 (NA)	0.79 (0.49 to 1.28)
Methionine	2 (0.0%)	0.69 (0.53 to 0.90)	1 (NA)	0.70 (0.39 to 1.28)		
Flavonoids	2 (0.0%)	1.02 (0.95 to 1.09)	2 (0.0%)	1.03 (0.93 to 1.14)		
Flavan-3-ols	1 (NA)	1.01 (0.70 to 1.46)	1 (NA)	0.89 (0.55 to 1.43)		
Flavones	1 (NA)	0.63 (0.41 to 0.96)	1 (NA)	0.48 (0.27 to 0.84)		
Flavonols	1 (NA)	1.12 (0.78 to 1.62)	1 (NA)	1.20 (0.77 to 1.87)		
Flavonones	1 (NA)	1.03 (0.72 to 1.48)	1 (NA)	0.98 (0.62 to 1.56)		
Phenolic acids	1 (NA)	0.97 (0.91 to 1.05)	1 (NA)	0.99 (0.89 to 1.10)		
Stilebenes	1 (NA)	0.97 (0.95 to 1.00)	1 (NA)	0.97 (0.94 to 1.00)		
Anthocyanidins	1 (NA)	1.42 (1.01 to 2.00)	1 (NA)	0.68 (0.41 to 1.13)		
Folate	5 (71.5%)	0.84 (0.65 to 1.07)	2 (68.0%)	0.96 (0.61 to 1.50)		
Isoflavones	7 (44.1%)	0.80 (0.69 to 0.93)	4 (0.0%)	0.89 (0.75 to 1.05)	4 (50.1%)	0.79 (0.65 to 0.97)
				1 (NA)	0.77 (0.60 to 0.98)	
Daidzein					1 (NA)	0.96 (0.52 to 1.76)
Genistein					1 (NA)	0.95 (0.52 to 1.75)
Glycetin					1 (NA)	0.80 (0.42 to 1.50)
Lignans	2 (5.1%)	0.96 (0.86 to 1.06)	2 (0.0%)	0.84 (0.73 to 0.96)		
Dietary patterns						
Healthy diet pattern	3 (4.90%)	0.76 (0.60 to 0.95)	3 (0.0%)	0.92 (0.70 to 1.22)	2 (0.30%)	0.82 (0.61 to 1.09)
Unhealthy diet pattern	4 (0.0%)	1.43 (1.17 to 1.76)	4 (0.0%)	1.03 (0.79 to 1.33)	2 (0.0%)	0.94 (0.70 to 1.27)
Dietary scores/indexes						
ACS guidelines	2 (0.0%)	0.98 (0.85 to 1.12)	2 (0.0%)	1.17 (0.88 to 1.54)		
CHFP	2 (0.0%)	0.70 (0.57 to 0.87)	2 (0.0%)	0.65 (0.50 to 0.86)		
DII	2 (55.5%)	1.14 (0.86 to 1.52)	2 (50.5%)	1.13 (0.76 to 1.67)		
HEI	4 (0.0%)	0.77 (0.64 to 0.91)	5 (53.3%)	1.01 (0.74 to 1.38)		
DASH	1 (NA)	0.66 (0.49 to 0.91)	2 (40.8%)	0.73 (0.52 to 1.02)		
aMED	1 (NA)	0.87 (0.64 to 1.17)	1 (NA)	1.15 (0.74 to 1.77)		
DQIR	1 (NA)	0.78 (0.58 to 1.07)	1 (NA)	0.81 (0.53 to 1.24)		
Glycemic index	1 (NA)	1.40 (0.78 to 2.50)	1 (NA)	1.60 (0.80 to 3.21)	1 (NA)	1.56 (0.77 to 3.31)
Glycemic load	1 (NA)	1.23 (0.46 to 3.31)	1 (NA)	1.11 (0.37 to 3.34)		1.14 (0.38 to 3.44)
NEAP	1 (NA)	1.54 (1.04 to 2.29)	1 (NA)	1.52 (1.01 to 2.32)	1 (NA)	1.15 (0.88 to 1.50)
PRAL	1 (NA)	1.30 (0.87 to 1.94)	1 (NA)	1.27 (0.83 to 1.94)	1 (NA)	1.09 (0.83 to 1.43)
RFS	1 (NA)	1.03 (0.74 to 1.42)	1 (NA)	1.54 (0.95 to 2.47)		
WCRF/AICR adherence score	1 (NA)	0.61 (0.39 to 0.96)	1 (NA)	0.88 (0.41 to 1.91)		
Prediagnostic	All-cause mortality		Breast-cancer-specific mortality		Breast cancer recurrence	
	N (I^2^)	RR/HR (95% CI)	N (I^2^)	RR/HR (95% CI)	N (I^2^)	RR/HR (95% CI)
Single food items						
Alcohol	6 (56.5%)	0.98 (0.82 to 1.17)	9 (60.1%)	1.17 (0.95 to 1.43)	2 (44.8%)	1.30 (0.86 to 1.97)
Beer	1 (NA)	0.92 (0.87 to 0.97)	2 (74.2%)	1.14 (0.70 to 1.87)	2 (0.0%)	1.42 (1.03 to 1.95)
Wine	1 (NA)	0.88 (0.84 to 0.93)	1 (NA)	0.98 (0.91 to 1.06)	1 (NA)	0.42 (0.18 to 0.95)
Spirits	1 (NA)	0.91 (0.87 to 0.96)	1 (NA)	0.92 (0.85 to 1.00)		
Tea	1 (NA)	0.94 (0.72 to 1.23)	1 (NA)	1.02 (0.67 to 1.55)		
Coffee	1 (NA)	1.12 (0.84 to 1.51)	1 (NA)	1.14 (0.71 to 1.83)		
Grains	1 (NA)	1.08 (0.91 to 1.29)	1 (NA)	1.07 (0.79 to 1.46)		
Whole grain products	2 (0.0%)	1.02 (0.96 to 1.08)	2 (0.0%)	1.05 (0.98 to 1.12)	2 (0.0%)	1.03 (0.97 to 1.11)
Oatmeal/muesli	1 (NA)	0.80 (0.61 to 1.05)	1 (NA)	0.99 (0.73 to 1.35)	1 (NA)	1.03 (0.83 to 1.29)
Rye bread	1 (NA)	1.10 (0.98 to 1.25)	1 (NA)	1.11 (0.96 to 1.29)	1 (NA)	1.04 (0.89 to 1.20)
Dairy products	2 (29.6%)	0.96 (0.75 to 1.25)	1 (NA)	0.98 (0.91 to 1.06)	1 (NA)	0.98 (0.91 to 1.06)
High-fat dairy products					1 (NA)	1.09 (0.88 to 1.35)
Low-fat dairy products					1 (NA)	0.84 (0.69 to 1.04)
Milk	1 (NA)	1.03 (0.96 to 1.10)	1 (NA)	0.99 (0.91 to 1.08)	1 (NA)	0.96 (0.89 to 1.05)
Cheese	1 (NA)	1.16 (0.97 to 1.17)	1 (NA)	1.18 (0.94 to 1.47)	1 (NA)	1.17 (0.94 to 1.45)
Butter/margarine/lard			1 (NA)	1.16 (0.86 to 1.58)	1 (NA)	1.30 (1.03 to 1.64)
Yogurt	1 (NA)	0.90 (0.75 to 1.07)	1 (NA)	0.86 (0.69 to 1.08)	1 (NA)	1.02 (0.83 to 1.27)
Soy products	1 (NA)	1.03 (0.81 to 1.33)	1 (NA)	1.03 (0.71 to 1.50)		
Fruits and vegetables	1 (NA)	1.06 (0.85 to 1.33)	1 (NA)	1.00 (0.66 to 1.50)		
Vegetables	1 (NA)	0.98 (0.62 to 1.53)				
Fish	1 (NA)	0.94 (0.62 to 1.43)			1 (NA)	0.93 (0.76 to 1.15)
Poultry	1 (NA)	0.60 (0.39 to 0.92)			1 (NA)	0.85 (0.69 to 1.05)
Red and processed meat	1 (NA)	0.88 (0.73 to 1.06)	2 (0.0%)	1.16 (0.87 to 1.54)	2 (0.0%)	1.04 (0.85 to 1.28)
Ginseng	1 (NA)	0.71 (0.52 to 0.98)			1 (NA)	0.70 (0.53 to 0.93)
Macronutrients						
Carbohydrates			2 (0.0%)	1.15 (0.71 to 1.87)		
E-Carb			1 (NA)	1.70 (0.70 to 3.80)		
Fat	1 (NA)	1.21 (0.78 to 1.90)	3 (25.1%)	1.12 (0.80 to 1.59)		
Saturated fat			3 (0.0%)	2.03 (1.26 to 3.27)		
Saturated fat/total fat			1 (NA)	1.93 (1.00 to 3.74)		
Monounsaturated fat			1 (NA)	1.33 (0.56 to 3.13)		
Polyunsaturated fat			1 (NA)	0.90 (0.52 to 1.55)		
18:2 trans fatty acid	1 (NA)	1.58 (1.03 to 2.43)				
Omega-3 fatty acids	1 (NA)	1.00 (0.62 to 1.60)				
Protein	1 (NA)	0.70 (0.46 to 1.08)	2 (23.7%)	0.53 (0.29 to 0.96)	1 (NA)	0.84 (0.69 to 1.03)
Soy protein					1 (NA)	0.99 (0.73 to 1.33)
Essential amino acids					1 (NA)	0.86 (0.71 to 1.05)
Animal protein					1 (NA)	0.74 (0.61 to 0.91)
Vegetable protein					1 (NA)	1.29 (1.05 to 1.59)
Branched-chain amino acids					1 (NA)	0.82 (0.68 to 1.00)
Fiber	1 (NA)	0.77 (0.45 to 1.25)	1 (NA)	0.70 (0.40 to 1.30)		
Micronutrients						
Vitamin A			1 (NA)	0.56 (0.28 to 1.09)		
Retinol			1 (NA)	1.05 (0.56 to 1.95)		
Vitamin B	1 (NA)	0.87 (0.74 to 1.02)				
Thiamin	1 (NA)	0.54 (0.38 to 0.88)	2 (0.0%)	0.48 (0.30 to 0.78)		
Riboflavin	1 (NA)	0.92 (0.58 to 1.44)	1 (NA)	0.72 (0.41 to 1.29)		
Niacin	1 (NA)	0.61 (0.38 to 0.98)	1 (NA)	0.61 (0.34 to 1.09)		
Vitamin B6	1 (NA)	0.95 (0.61 to 1.48)	1 (NA)	0.77 (0.44 to 1.36)		
Vitamin B12	1 (NA)	1.20 (0.80 to 1.81)	1 (NA)	1.10 (0.65 to 1.85)		
Vitamin C	1 (NA)	0.84 (0.71 to 1.00)	3 (12.3%)	0.71 (0.54 to 0.92)		
Vitamin D	1 (NA)	0.78 (0.70 to 0.88)				
Vitamin E			1 (NA)	0.55 (0.26 to 1.17)		
Beta-Carotene			2 (52.2%)	0.70 (0.36 to 1.38)		
Provitamin A	1 (NA)	0.58 (0.34 to 0.99)				
Lutein/zeaxanthin	1 (NA)	0.85 (0.53 to 1.38)				
Calcium	2 (0.0%)	0.71 (0.51 to 1.00)	2 (43.5%)	0.92 (0.47 to 1.79)		
Magnesium	1 (NA)	0.50 (0.28 to 0.90)	1 (NA)	0.70 (0.33 to 1.49)		
Betaine	1 (NA)	0.81 (0.54 to 1.20)	1 (NA)	0.72 (0.44 to 1.17)		
Methionine	1 (NA)	0.70 (0.44 to 1.13)	1 (NA)	0.70 (0.39 to 1.28)		
Flavonoids	2 (0.0%)	1.02 (0.95 to 1.09)	2 (0.0%)	1.03 (0.93 to 1.14)		
Flavan-3-ols	1 (NA)	1.01 (0.70 to 1.46)	1 (NA)	0.89 (0.55 to 1.43)		
Flavonols	1 (NA)	1.12 (0.78 to 1.62)	1 (NA)	1.20 (0.77 to 1.87)		
Flavonones	1 (NA)	1.03 (0.72 to 1.48)	1 (NA)	0.98 (0.62 to 1.56)		
Flavones	1 (NA)	0.63 (0.41 to 0.96)	1 (NA)	0.48 (0.27 to 0.84)		
Phenolic acids			1 (NA)	0.99 (0.89 to 1.10)		
Stilebenes			1 (NA)	0.97 (0.94 to 1.00)		
Anthocyanidins	1 (NA)	1.42 (1.01 to 2.00)	1 (NA)	0.68 (0.41 to 1.13)		
Folate	2 (18.3%)	0.83 (0.68 to 1.00)	2 (68.0%)	0.96 (0.61 to 1.50)		
Isoflavone	3 (70.4%)	0.81 (0.60 to 1.08)	3 (0.0%)	0.93 (0.74 to 1.17)	2 (82.8%)	0.83 (0.53 to 1.31)
Daidzein					1 (NA)	0.96 (0.52 to 1.76)
Genistein					1 (NA)	0.95 (0.52 to 1.75)
Glycetin					1 (NA)	0.80 (0.42 to 1.50)
Lignans			2 (0.0%)	0.84 (0.73 to 0.96)		
Dietary patterns						
Healthy diet pattern	2 (51.5%)	0.72 (0.48 to 1.09)	2 (0.0%)	0.86 (0.61 to 1.20)	2 (0.30%)	0.82 (0.61 to 1.09)
Unhealthy diet pattern	3 (0.0%)	1.40 (1.10 to 1.78)	3 (0.0%)	1.04 (0.77 to 1.40)	2 (0.0%)	0.94 (0.70 to 1.27)
Dietary scores/indexes						
ACS guidelines	1 (NA)	1.00 (0.84 to 1.18)	1 (NA)	1.06 (0.79 to 1.42)		
DII	1 (NA)	1.00 (0.78 to 1.28)	1 (NA)	0.97 (0.73 to 1.27)		
Postdiagnostic	All-cause mortality		Breast-cancer-specific mortality		Breast cancer recurrence	
	N (I^2^)	RR/HR (95% CI)	N (I^2^)	RR/HR (95% CI)	N (I^2^)	RR/HR (95% CI)
Single food items						
Alcohol	4 (43.3%)	0.88 (0.75 to 1.03)	4 (57.9%)	1.04 (0.77 to 1.41)	2 (46.2%)	1.16 (0.90 to 1.49)
Beer	1 (NA)	1.02 (0.85 to 1.23)	1 (NA)	1.26 (0.88 to 1.79)		
Wine	2 (0.0%)	0.93 (0.80 to 1.09)	2 (15.3%)	1.14 (0.84 to 1.54)	1 (NA)	1.33 (0.97 to 1.81)
Spirits	1 (NA)	0.84 (0.70 to 1.00)	1 (NA)	0.74 (0.53 to 1.03)		
Tea	1 (NA)	0.58 (0.29 to 1.16)	1 (NA)		0.60 (0.29 to 1.27)	
Grains	1 (NA)	1.09 (0.86 to 1.38)	1 (NA)	1.24 (0.81 to 1.88)		
Whole grain products	2 (0.0%)	0.97 (0.89 to 1.07)	2 (0.0%)	1.02 (0.91 to 1.13)	2 (26.2%)	0.92 (0.78 to 1.08)
Oatmeal/muesli	1 (NA)	0.75 (0.53 to 1.07)	1 (NA)	0.82 (0.55 to 1.22)	1 (NA)	0.93 (0.62 to 1.40)
Rye bread	1 (NA)	1.21 (0.98 to 1.47)	1 (NA)	1.34 (1.06 to 1.70)	1 (NA)	1.27 (0.97 to 1.66)
Dairy products	2 (31.3%)	1.03 (0.89 to 1.19)	2 (0.0%)	0.99 (0.87 to 1.12)	1 (NA)	0.93 (0.80 to 1.07)
Milk	1 (NA)	1.00 (0.90 to 1.12)	1 (NA)	1.02 (0.89 to 1.17)	1 (NA)	0.92 (0.78 to 1.08)
Cheese	1 (NA)	1.09 (0.70 to 1.47)	1 (NA)	0.95 (0.66 to 1.37)	1 (NA)	1.23 (0.85 to 1.78)
Yogurt	1 (NA)	0.93 (0.74 to 1.16)	1 (NA)	0.84 (0.60 to 1.18)	1 (NA)	0.92 (0.67 to 1.25)
Any fruits, fruit juices, and vegetables	1 (NA)	0.68 (0.42 to 1.09)				
Fruits and vegetables	1 (NA)	1.03 (0.80 to 1.33)	1 (NA)	1.31 (0.83 to 2.06)		
Fruits	2 (80.4%)	0.94 (0.44 to 2.23)	1 (NA)	1.39 (0.64 to 2.99)		
Fruits and fruit juices	1 (NA)	0.87 (0.57 to 1.35)				
Citrus fruits	1 (NA)	0.93 (0.61 to 1.42)				
Vegetables	3 (73.6%)	0.97 (0.58 to 1.63)	1 (NA)	0.96 (0.38 to 2.45)		
Cruciferous vegetables	2 (0.0%)	1.03 (0.83 to 1.28)	1 (NA)	0.95 (0.59 to 1.54)		
Leafy vegetables	1 (NA)	0.72 (0.41 to 1.24)				
Yellow vegetables	1 (NA)	0.90 (0.58 to 1.40)				
Red and processed meat	2 (86.5%)	0.84 (0.49 to 1.46)	2 (0.0%)	0.88 (0.61 to 1.28)		
Natural products	1 (NA)	0.95 (0.67 to 1.35)	1 (NA)	1.15 (0.69 to 1.94)		
Macronutrients						
Carbohydrates	2 (0.0%)	0.97 (0.73 to 1.29)	2 (0.0%)	0.89 (0.55 to 1.45)	1 (NA)	0.77 (0.27 to 2.19)
Fat	2 (91.5%)	1.76 (0.61 to 5.11)	1 (NA)	0.92 (0.53 to 1.60)		
Trans fat	1 (NA)	1.78 (1.35 to 2.32)	1 (NA)	1.42 (0.80 to 2.52)		
Saturated fat	2 (89.3%)	2.40 (0.78 to 7.38)	1 (NA)	1.55 (0.88 to 2.75)		
Monounsaturated fat	1 (NA)	1.14 (0.86 to 1.52)	1 (NA)	0.89 (0.49 to 1.60)		
Polyunsaturated fat	1 (NA)	0.91 (0.70 to 1.19)	1 (NA)	1.63 (0.87 to 3.03)		
Linoleic fatty acids	1 (NA)	2.39 (1.21 to 4.69)				
Oleic fatty acids	1 (NA)	3.56 (1.67 to 7.59)				
Protein	1 (NA)	0.98 (0.73 to 1.31)	1 (NA)	1.19 (0.66 to 2.14)		
Soy protein	1 (NA)	0.71 (0.54 to 0.92)	1 (NA)		0.68 (0.54 to 0.87)	
Fiber	3 (0.0%)	0.64 (0.48 to 0.86)	3 (0.0%)	0.77 (0.52 to 1.15)	1 (NA)	0.68 (0.27 to 1.70)
Micronutrients						
Antioxidants	2 (0.0%)	0.83 (0.73 to 0.95)	1 (NA)	0.86 (0.73 to 1.02)	2 (57.3%)	0.87 (0.73 to 1.04)
				1 (NA)	0.54 (0.27 to 1.04)	
Multivitamins	4 (0.0%)	0.88 (0.79 to 0.99)	3 (0.0%)	0.89 (0.79 to 1.02)	4 (0.0%)	0.90 (0.82 to 0.99)
Vitamin A	2 (0.0%)	1.05 (0.89 to 1.26)	2 (0.0%)	1.01 (0.77 to 1.32)	1 (NA)	1.07 (0.72 to 1.60)
Retinol	1 (NA)	1.05 (0.86 to 1.30)				
Vitamin B			1 (NA)	0.90 (0.74 to 1.10)	1 (NA)	0.88 (0.75 to 1.03)
Niacin	1 (NA)	0.80 (0.52 to 1.51)				
Vitamin B6	1 (NA)	1.02 (0.74 to 1.42)				
Vitamin C	6 (58.9%)	0.85 (0.71 to 1.02)	2 (0.0%)	0.86 (0.75 to 1.01)	3 (28.5%)	0.84 (0.73 to 0.96)
				1 (NA)	0.64 (0.32 to 1.27)	
Vitamin D	4 (0.0%)	0.86 (0.78 to 0.95)	3 (0.0%)	0.85 (0.72 to 1.01)	1 (NA)	0.82 (0.60 to 1.12)
Vitamin E	5 (0.0%)	0.82 (0.73 to 0.92)	2 (0.0%)	0.86 (0.71 to 1.03)	3 (51.4%)	0.78 (0.64 to 0.95)
				1 (NA)	0.55 (0.28 to 1.08)	
Carotenoids	1 (NA)	1.63 (1.06 to 2.50)	1 (NA)	1.93 (1.14 to 3.28)	1 (NA)	1.23 (0.76 to 1.96)
Alpha-Carotene	3 (0.0%)	1.05 (0.85 to 1.30)	1 (NA)	0.98 (0.59 to 1.64)		
Beta-Carotene	4 (62.4%)	0.96 (0.68 to 1.36)	2 (0.0%)	1.16 (0.76 to 1.78)	1 (NA)	0.89 (0.50 to 1.60)
Beta-Cryptoxanthin	3 (72.5%)	0.86 (0.54 to 1.38)	1 (NA)	0.81 (0.45 to 1.45)		
Lutein/zeaxanthin	3(69.5%)	0.74 (0.47 to 1.16)	1 (NA)	1.16 (0.62 to 2.19)		
Lycopene	4 (0.0%)	0.98 (0.78 to 1.22)	2 (0.0%)	1.51 (0.90 to 2.55)	1 (NA)	1.17 (0.35 to 3.89)
Selenium	2 (0.0%)	0.93 (0.59 to 1.46)	1 (NA)	0.90 (0.45 to 1.79)	1 (NA)	0.89 (0.53 to 1.49)
Calcium	2 (0.0%)	0.85 (0.71 to 1.01)	1 (NA)	0.59 (0.32 to 1.08)		
Magnesium	1 (NA)	1.02 (0.68 to 1.53)				
Zinc	2 (44.7%)	0.96 (0.67 to 1.37)	1 (NA)	0.82 (0.44 to 1.53)	1 (NA)	0.79 (0.49 to 1.28)
Folate	2 (91.0%)	0.64 (0.20 to 2.01)				
Iron	1 (NA)	1.60 (0.91 to 2.90)				
Isoflavone	4 (2.2%)	0.79 (0.67 to 0.93)	1 (NA)	1.03 (0.46 to 2.28)	2 (0.0%)	0.75 (0.62 to 0.91)
				1 (NA)	0.77 (0.60 to 0.98)	
Dietary patterns						
Healthy diet pattern	1 (NA)	0.78 (0.54 to 1.12)	1 (NA)	1.07 (0.66 to 1.73)		
Unhealthy diet pattern	1 (NA)	1.53 (1.03 to 2.29)	1 (NA)	1.01 (0.60 to 1.70)		
Dietary scores/indexes						
ACS guidelines	1 (NA)	0.93 (0.73 to 1.18)	1 (NA)	1.44 (0.90 to 2.30)		
CHFP	2 (0.0%)	0.70 (0.57 to 0.87)	2 (0.0%)	0.65 (0.50 to 0.86)		
DII	1 (NA)	1.34 (1.01 to 1.81)	1 (NA)	1.47 (0.89 to 2.43)		
HEI	3 (24.3%)	0.77 (0.59 to 1.00)	3 (75.3%)	0.90 (0.43 to 1.88)		
DASH	1 (NA)	0.66 (0.49 to 0.91)	1 (NA)	0.85 (0.61 to 1.19)		
aMED	1 (NA)	0.87 (0.64 to 1.17)	1 (NA)	1.15 (0.74 to 1.77)		
DQIR	1 (NA)	0.78 (0.58 to 1.07)	1 (NA)	0.81 (0.53 to 1.24)		
Glycemic index	1 (NA)	1.40 (0.78 to 2.50)	1 (NA)	1.60 (0.80 to 3.21)	1 (NA)	1.56 (0.77 to 3.31)
Glycemic load	1 (NA)	1.23 (0.46 to 3.31)	1 (NA)	1.11 (0.37 to 3.34)	1 (NA)	1.14 (0.38 to 3.44)
RFS	1 (NA)	1.03 (0.74 to 1.42)				

Abbreviations: ACS, American Cancer Society; CHFP, Chinese Food Pagoda; DII, Dietary Inflammatory Index; HEI, Health Eating Index; DASH, Dietary Approaches to Stop Hypertension; aMED, Alternate Mediterranean Score; DQIR, Diet Quality Index—Revised; NEAP, the net endogenous acid production; PRAL, the potential renal acid load; RFS, recommended food score; WCRF/AICR, World Cancer Research Fund and American Institute for Cancer Research; NA, not applicable.

**Table 2 cancers-13-05329-t002:** Systematic review and meta-analysis of associations between dietary factors and breast cancer prognosis among breast cancer survivors by menopausal status.

	All-Cause Mortality		Breast-Cancer-Specific Mortality		Breast Cancer Recurrence	
Premenopause	N (I^2^)	RR/HR (95% CI)	N (I^2^)	RR/HR (95% CI)	N (I^2^)	RR/HR (95% CI)
Single food items						
Alcohol	1 (NA)	0.23 (0.03 to 1.54)				
Beer			1 (NA)	2.33 (1.35 to 4.00)	1 (NA)	1.58 (1.15 to 2.17)
Butter/margarine/lard			1 (NA)	1.03 (0.61 to 1.76)	1 (NA)	1.67 (1.17 to 2.39)
Any fruits, fruit juices, and vegetables	1 (NA)	1.38 (0.65 to 2.91)				
Fruits and fruit juices	1 (NA)	1.10 (0.48 to 2.52)				
Citrus fruits	1 (NA)	1.70 (0.75 to 3.89)				
Vegetables	1 (NA)	1.40 (0.71 to 2.76)				
Cruciferous vegetables	1 (NA)	0.72 (0.34 to 1.54)				
Leafy vegetables	1 (NA)	0.85 (0.39 to 1.85)				
Yellow vegetables	1 (NA)	1.09 (0.52 to 2.28)				
Meat/liver/bacon			1 (NA)	2.60 (0.96 to 7.03)	1 (NA)	1.93 (0.89 to 4.15)
Macronutrients						
Carbohydrates			1 (NA)	1.30 (0.30 to 5.10)		
E-Carb			1 (NA)	2.10 (0.50 to 8.60)		
Fat			1 (NA)	4.80 (1.30 to 18.10)		
Saturated fat			2 (78.8%)	2.09 (0.51 to 8.54)		
Saturated fat/total fat			1 (NA)	1.25 (0.47 to 3.31)		
Protein			1 (NA)	0.20 (0.10 to 0.90)		
Soy protein					1 (NA)	1.09 (0.74 to 1.60)
Fiber			1 (NA)	0.70 (0.20 to 1.60)		
Micronutrients						
Vitamin A			1 (NA)	0.78 (0.54 to 1.17)		
Thiamin			1 (NA)	1.20 (1.07 to 3.66)		
Vitamin C	1 (NA)	0.90 (0.42 to 1.94)	1 (NA)	0.54 (0.30 to 0.98)		
Vitamin E	1 (NA)	0.96 (0.44 to 2.09)	1 (NA)	0.64 (0.33 to 1.26)		
Alpha-Carotene	1 (NA)	0.76 (0.35 to 1.67)				
Beta-Carotene	1 (NA)	0.82 (0.37 to 1.82)	1 (NA)	0.87 (0.65 to 1.18)		
Beta-Cryptoxanthin	1 (NA)	1.13 (0.53 to 2.41)				
Lutein/zeaxanthin	1 (NA)	1.71 (0.89 to 3.29)				
Lycopene	1 (NA)	0.61 (0.29 to 1.29)				
Flavonoids	2 (69.4%)	1.18 (0.65 to 2.15)	2 (45.0%)	1.18 (0.74 to 1.87)		
Flavan-3-ols	1 (NA)	1.76 (0.91 to 3.42)	1 (NA)	1.75 (0.83 to 3.69)		
Flavones	1 (NA)	0.69 (0.32 to 1.47)	1 (NA)	0.45 (0.17 to 1.19)		
Flavonols	1 (NA)	1.64 (0.84 to 3.17)	1 (NA)	1.64 (0.78 to 3.46)		
Flavonones	1 (NA)	1.08 (0.48 to 2.43)	1 (NA)	0.61 (0.21 to 1.81)		
Phenolic acids	1 (NA)	1.04 (0.90 to 1.20)	1 (NA)	1.00 (0.83 to 1.21)		
Stilebenes	1 (NA)	0.99 (0.92 to 1.06)	1 (NA)	0.99 (0.91 to 1.08)		
Anthocyanidins	1 (NA)	0.62 (0.27 to 1.40)	1 (NA)	0.81 (0.35 to 1.89)		
Calcium			1 (NA)	0.93 (0.58 to 1.50)		
Isoflavone	1 (NA)	0.71 (0.34 to 1.48)	2 (0.0%)	0.96 (0.69 to 1.34)	1 (NA)	0.88 (0.61 to 1.23)
Daidzein					1 (NA)	1.74 (0.63 to 4.76)
Genistein					1 (NA)	1.75 (0.65 to 4.76)
Glycetin					1 (NA)	1.60 (0.54 to 4.72)
Lignans	2 (0.0%)	1.26 (1.06 to 1.50)	2 (0.0%)	1.23 (0.98 to 1.55)		
Dietary scores/indexes						
DII	1 (NA)	0.84 (0.52 to 1.34)	1 (NA)	0.76 (0.46 to 1.26)		
	All-cause mortality	Breast cancer-specific mortality	Breast cancer recurrence			
Postmenopause	N (I^2^)	RR/HR (95% CI)	N (I^2^)	RR/HR (95% CI)	N (I^2^)	RR/HR (95% CI)
Single food items						
Alcohol	2 (9.2%)	1.14 (0.85 to 1.54)	1 (NA)	1.74 (1.13 to 2.67)	1 (NA)	1.08 (0.73 to 1.58)
Whole grain products	4 (0.0%)	1.01 (0.96 to 1.06)	4 (0.0%)	1.04 (0.98 to 1.10)	4 (14.0%)	1.01 (0.94 to 1.08)
Rye bread	2 (0.0%)	1.13 (1.02 to 1.25)	2 (43.0%)	1.19 (0.10 to 1.42)	2 (38.4%)	1.11 (0.92 to 1.34)
Oatmeal/muesli	2 (0.0%)	0.78 (0.63 to 0.97)	2 (0.0%)	0.92 (0.72 to 1.18)	2 (0.0%)	0.91 (0.71 to 1.16)
Dairy products	2 (0.0%)	1.01 (0.96 to 1.08)	2 (0.0%)	0.98 (0.92 to 1.05)	2 (0.0%)	0.97 (0.91 to 1.04)
Cheese	2 (0.0%)	1.16 (1.05 to 1.27)	2 (0.0%)	1.11 (0.92 to 1.35)	2 (0.0%)	1.18 (0.98 to 1.43)
Milk	2 (0.0%)	1.02 (0.96 to 1.08)	2 (0.0%)	1.00 (0.93 to 1.07)	2 (0.0%)	0.95 (0.88 to 1.02)
Yogurt	2 (0.0%)	0.91 (0.79 to 1.05)	2 (0.0%)	0.85 (0.71 to 1.03)	2 (0.0%)	0.99 (0.83 to 1.18)
Soy products	1 (NA)	1.03 (0.81 to 1.33)	1 (NA)	1.03 (0.71 to 1.50)		
Any fruits, fruit juices, and vegetables	1 (NA)	0.68 (0.42 to 1.09)				
Fruits	1 (NA)	0.63 (0.38 to 1.05)				
Fruits and fruit juices	1 (NA)	0.87 (0.57 to 1.35)				
Citrus fruits	1 (NA)	0.93 (0.61 to 1.42)				
Vegetables	2 (73.7%)	0.80 (0.42 to 1.51)				
Cruciferous vegetables	1 (NA)	1.07 (0.67 to 1.72)				
Leafy vegetables	1 (NA)	0.72 (0.41 to 1.24)				
Yellow vegetables	1 (NA)	0.90 (0.58 to 1.40)				
Macronutrients						
Carbohydrates			1 (NA)	2.00 (0.70 to 5.70)		
E-Carb			1 (NA)	1.70 (0.60 to 4.90)		
Fat	1 (NA)	3.12 (1.79 to 5.44)	1 (NA)	0.70 (0.20 to 2.20)		
Saturated fat	1 (NA)	4.45 (2.26 to 8.78)	2 (0.0%)	1.29 (0.96 to 1.72)		
Saturated fat/total fat			1 (NA)	2.53 (1.20 to 5.33)		
Linoleic fatty acids	1 (NA)	2.39 (1.21 to 4.69)				
Oleic fatty acids	1 (NA)	3.56 (1.67 to 7.59)				
Fiber	1 (NA)	0.48 (0.27 to 0.86)	1 (NA)	0.80 (0.30 to 1.80)		
Protein			1 (NA)	0.60 (0.20 to 1.60)		
Soy protein					1 (NA)	0.79 (0.49 to 1.28)
Micronutrients						
Antioxidants			1 (NA)		0.54 (0.27 to 1.04)	
Vitamin A			1 (NA)	0.84 (0.67 to 1.06)		
Thiamin			1 (NA)	0.62 (0.36 to 1.05)		
Vitamin C	2 (82.7%)	0.71 (0.30 to 1.68)	1 (NA)	0.74 (0.50 to 1.11)		
			1 (NA)		0.64 (0.32 to 1.27)	
Vitamin D	1 (NA)	0.78 (0.70 to 0.88)				
Vitamin E	1 (NA)	0.77 (0.47 to 1.27)	1 (NA)	0.76 (0.51 to 1.13)		
			1 (NA)		0.55 (0.28 to 1.08)	
Alpha-Carotene	2 (40.1%)	0.99 (0.65 to 1.53)				
Beta-Carotene	2 (78.8%)	0.74 (0.35 to 1.57)	1 (NA)	0.84 (0.68 to 1.03)		
Beta-Cryptoxanthin	2 (20.5%)	0.70 (0.47 to 1.04)				
Provitamin A	1 (NA)	0.58 (0.34 to 0.99)				
Lutein/zeaxanthin	2 (0.0%)	0.59 (0.41 to 0.84)				
Lycopene	2 (0.0%)	0.78 (0.54 to 1.11)				
Flavonoids	2 (18.7%)	0.99 (0.84 to 1.17)	2 (60.5%)	0.88 (0.55 to 1.41)		
Flavan-3-ols	1 (NA)	0.84 (0.53 to 1.32)	1 (NA)	0.63 (0.34 to 1.18)		
Flavones	1 (NA)	0.59 (0.35 to 0.99)	1 (NA)	0.49 (0.24 to 0.99)		
Flavonols	1 (NA)	0.98 (0.62 to 1.53)	1 (NA)	1.02 (0.59 to 1.79)		
Flavonones	1 (NA)	0.99 (0.66 to 1.49)	1 (NA)	1.09 (0.65 to 1.82)		
Phenolic acids	1 (NA)	0.97 (0.91 to 1.05)	1 (NA)	0.99 (0.89 to 1.10)		
Stilebenes	1 (NA)	0.97 (0.95 to 1.00)	1 (NA)	0.97 (0.94 to 1.00)		
Anthocyanidins	1 (NA)	0.66 (0.40 to 1.08)	1 (NA)	0.62 (0.33 to 1.18)		
Isoflavone	2 (83.9%)	0.70 (0.32 to 1.53)	3 (0.0%)	0.91 (0.74 to 1.13)	1 (NA)	0.67 (0.58 to 0.92)
Daidzein					1 (NA)	0.70 (0.27 to 1.77)
Genistein					1 (NA)	0.69 (0.27 to 1.75)
Glycetin					1 (NA)	0.51 (0.18 to 1.38)
Lignans	2 (0.0%)	0.94 (0.86 to 1.03)	2 (0.0%)	0.83 (0.72 to 0.96)		
Calcium			1 (NA)	0.71 (0.49 to 1.05)		
Folate	1 (NA)	0.34 (0.18 to 0.67)				
Dietary patterns						
Healthy diet pattern	1 (NA)	0.87 (0.61 to 1.23)	1 (NA)	0.89 (0.59 to 1.35)	1 (NA)	0.71 (0.48 to 1.06)
Unhealthy diet pattern	1 (NA)	1.34 (0.93 to 1.94)	1 (NA)	0.99 (0.64 to 1.52)	1 (NA)	0.91 (0.61 to 1.36)
Dietary scores/indexes						
DII	2 (25.3%)	1.19 (0.93 to 1.51)	2 (5.6%)	1.18 (0.89 to 1.58)		
HEI	1 (NA)	0.74 (0.55 to 0.99)	1 (NA)	0.91 (0.60 to 1.40)		
WCRF/AICR adherence score	1 (NA)	0.61 (0.39 to 0.96)	1 (NA)	0.88 (0.41 to 1.91)		

Abbreviations: DII, Dietary Inflammatory Index; HEI, Health Eating Index; WCRF/AICR, World Cancer Research Fund/American Institute for Cancer Research; NA, not applicable.

**Table 3 cancers-13-05329-t003:** Systematic review and meta-analysis of associations between dietary or supplementary micronutrient intake and breast cancer prognosis among breast cancer survivors.

	All-Cause Mortality		Breast-Cancer-Specific Mortality		Breast Cancer Recurrence	
Dietary	N (I^2^)	RR/HR (95% CI)	N (I^2^)	RR/HR (95% CI)	N (I^2^)	RR/HR (95% CI)
Vitamin A	2 (65.2%)	0.94 (0.66 to 1.34)	1 (NA)	1.24 (0.68 to 2.24)		
Retinol			1 (NA)	1.05 (0.56 to 1.95)		
Thiamin	1 (NA)	0.54 (0.38 to 0.88)	1 (NA)	0.44 (0.24 to 0.81)		
Riboflavin	1 (NA)	0.92 (0.58 to 1.44)	1 (NA)	0.72 (0.41 to 1.29)		
Niacin	2 (0.0%)	0.74 (0.57 to 0.96)	1 (NA)	0.61 (0.34 to 1.09)		
Vitamin B6	1 (NA)	0.95 (0.61 to 1.48)	1 (NA)	0.77 (0.44 to 1.36)		
Vitamin B12	1 (NA)	1.20 (0.80 to 1.81)	1 (NA)	1.10 (0.65 to 1.85)		
Vitamin C	3 (66.1%)	0.78 (0.54 to 1.14)	2 (0.0%)	0.76 (0.59 to 0.97)		
Vitamin D	1 (NA)	0.86 (0.64 to 1.16)	1 (NA)	1.02 (0.58 to 1.79)		
Vitamin E	1 (NA)	0.77 ().47 to 1.27)				
Carotenes	1 (NA)	0.96 (0.70 to 1.31)				
Alpha-Carotene	4 (0.0%)	1.01 (0.85 to 1.21)	1 (NA)	0.98 (0.59 to 1.64)		
Beta-Carotene	4 (61.4%)	0.92 (0.68 to 1.24)	3 (34.5%)	0.82 (0.53 to 1.29)		
Beta-Cryptoxanthin	4 (61.2%)	0.88 (0.64 to 1.20)	1 (NA)	0.81 (0.45 to 1.45)		
Lutein/zeaxanthin	4 (54.9%)	0.79 (0.58 to 1.06)	1 (NA)	1.16 (0.62 to 2.19)		
Lycopene	4 (11.7%)	0.90 (0.74 to 1.09)	1 (NA)	1.42 (0.80 to 2.50)		
Selenium	1 (NA)	0.86 (0.63 to 1.19)				
Betaine	1 (NA)	0.81 (0.54 to 1.20)	1 (NA)	0.72 (0.44 to 1.17)		
Calcium	3 (0.0%)	0.72 (0.58 to 0.89)	2 (62.5%)	0.85 (0.39 to 1.86)		
Magnesium	2 (24.1%)	0.66 (0.46 to 0.94)	1 (NA)	0.70 (0.33 to 1.49)		
Methionine	2 (0.0%)	0.69 (0.53 to 0.90)	1 (NA)	0.70 (0.39 to 1.28)		
Iodine	1 (NA)	0.90 (0.67 to 1.21)				
Potassium	1 (NA)	0.98 (0.69 to 1.38)				
Sodium	1 (NA)	0.79 (0.57 to 1.09)				
Flavonoids	1 (NA)	1.02 (0.95 to 1.10)	1 (NA)	1.04 (0.93 to 1.15)		
Phenolic acids	1 (NA)	0.97 (0.91 to 1.05)	1 (NA)	0.99 (0.89 to 1.10)		
Stilebenes	1 (NA)	0.97 (0.95 to 1.00)	1 (NA)	0.97 (0.94 to 1.00)		
Folate	3 (65.5%)	0.66 (0.46 to 0.96)	2 (0.0%)	0.79 (0.61 to 1.01)		
Isoflavone	3 (0.0%)	0.90 (0.79 to 1.02)	3 (0.0%)	0.91 (0.77 to 1.08)	2 (0.0%)	0.78 (0.65 to 0.93)
			1 (NA)		0.77 (0.60 to 0.98)	
Daidzein					1 (NA)	0.96 (0.52 to 1.76)
Genistein					1 (NA)	0.95 (0.52 to 1.75)
Glycetin					1 (NA)	0.80 (0.42 to 1.50)
Lignans	1 (NA)	0.94 (0.86 to 1.04)	1 (NA)	0.83 (0.72 to 0.96)		
Caffeine	1 (NA)	0.77 (0.55 to 1.07)				
	All-cause mortality		Breast-cancer-specific mortality		Breast cancer recurrence	
Supplementary	N (I^2^)	RR/HR (95% CI)	N (I^2^)	RR/HR (95% CI)	N (I^2^)	RR/HR (95% CI)
Antioxidants	2 (0.0%)	0.83 (0.73 to 0.95)	1 (NA)	0.86 (0.73 to 1.02)	2 (57.3%)	0.87 (0.73 to 1.04)
				1 (NA)	0.54 (0.27 to 1.04)	
Multivitamins	4 (0.0%)	0.88 (0.79 to 0.99)	3 (0.0%)	0.89 (0.79 to 1.02)	4 (0.0%)	0.90 (0.82 to 0.99)
Vitamin A	1 (NA)	1.02 (0.82 to 1.27)	2 (50.4%)	0.80 (0.49 to 1.32)	1 (NA)	1.07 (0.72 to 1.60)
Retinol	2 (36.9%)	0.96 (0.76 to 1.21)				
Vitamin B	1 (NA)	0.87 (0.74 to 1.02)	1 (NA)	0.90 (0.74 to 1.10)	1 (NA)	0.88 (0.75 to 1.03)
Thiamin	1 (NA)	0.82 (0.59 to 1.13)	1 (NA)	0.57 (0.26 to 1.25)		
Riboflavin	1 (NA)	0.81 (0.58 to 1.13)				
Niacin	1 (NA)	0.80 (0.52 to 1.51)				
Vitamin B6	2 (21.5%)	0.89 (0.69 to 1.16)				
Vitamin C	6 (57.5%)	0.89 (0.75 to 1.06)	3 (45.9%)	0.79 (0.61 to 1.02)	3 (28.5%)	0.84 (0.73 to 0.96)
				1 (NA)	0.64 (0.32 to 1.27)	
Vitamin D	5 (0.0%)	0.82 (0.76 to 0.89)	2 (0.0%)	0.84 (0.70 to 1.00)	1 (NA)	0.82 (0.60 to 1.12)
Vitamin E	5 (0.0%)	0.82 (0.73 to 0.92)	3 (0.0%)	0.84 (0.70 to 1.00)	3 (51.4%)	0.78 (0.64 to 0.95)
				1 (NA)	0.55 (0.28 to 1.08)	
Carotenoids	1 (NA)	1.63 (1.06 to 2.50)	1 (NA)	1.93 (1.14 to 3.28)	1 (NA)	1.23 (0.76 to 1.96)
Beta-Carotene	2 (49.2%)	0.91 (0.54 to 1.54)	1 (NA)	1.33 (0.69 to 2.55)	1 (NA)	0.89 (0.50 to 1.60)
Provitamin A	1 (NA)	0.58 (0.34 to 0.99)				
Lycopene	1 (NA)	1.38 (0.41 to 4.61)	1 (NA)	2.09 (0.59 to 7.43)	1 (NA)	1.17 (0.35 to 3.89)
Selenium	2 (0.0%)	0.93 (0.59 to 1.46)	1 (NA)	0.90 (0.45 to 1.79)	1 (NA)	0.89 (0.53 to 1.49)
Calcium	1 (NA)	0.90 (0.74 to 1.12)	1 (NA)	0.66 (0.33 to 1.31)		
Magnesium	1 (NA)	1.02 (0.68 to 1.53)				
Folate	3 (0.0%)	1.03 (0.86 to 1.22)				
Iron	2 (70.7%)	1.12 (0.61 to 2.03)				
Zinc	3 (24.8%)	0.92 (0.73 to 1.15)	1 (NA)	0.82 (0.44 to 1.53)	1 (NA)	0.79 (0.49 to 1.28)

Abbreviation: NA, not applicable.

## Data Availability

The data presented in this study are openly available at https://www.mdpi.com/journal/cancers.

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
