# Peer review of "Dietary Factors and Breast Cancer Prognosis among Breast Cancer Survivors: A Systematic Review and Meta-Analysis of Cohort Studies"

_cancers, 2021, doi:10.3390/cancers13215329_

Round 1

Reviewer 1 Report

This is a well written extensive systematic review and meta-analyses of cohort studies on dietary factors and breast cancer prognosis. In my opinion, it adds to the evidence provided bij the CUP (continuous update project) of the WCRF.

Remarks:

  • The influence of confounding factors is rarely discussed in the paper. The authors should elaborate on this. How was the confounding adjustment in the cohort studies? Did the studies adjust for example for physical activity? Is confounding incorporated in the quality adjustment? In the discussion section, the authors should discuss whether (residual confounding) could have influenced the results.
  • The authors describe in the introduction (line 59/60) and discussion (line 356/357) sections that RCTs might easily lead to selection bias. I am not convinced of this statement. RCTs might another way to test mechanisms and effects of specific diets or dietary components and for example supplements. A RCT can lead to selection bias by selective loss to follow-up. but is that what the authors mean? Or do they mean RCTs include selective populations with a healthier background diet??

In the same sentences, the authors state that case-control studies belong to the experimental study designs. This is not correct and should be adjusted.

  • The NOS should include a clear legend; what do all items and scores mean? Why are studies with scores below 6 considered as low-quality studies. What is the evidence for this cut-off?
  • 2.5 statistical analysis. Is it possible to add dose-response analyses?
  • 3.2 study characteristics (line 158/159)…> 120,000 women in total & 9,659 pre-menopausal and 35,366 postmenopausal women -> is the menopausal status of the remaining women unclear? Please explain.
  • 3.2 study characteristics (line 159/160): The total number of deaths is lower than the BC-specific deaths?? Please adjust.
  • Line 270: the intake of Rye Bread appeared positively associated with all-cause mortality in postmenopausal women, while this is a healthy product with a high fiber load. Can the authors explain this association? Chance finding? Confounding?
  • Line 303 en 304; VIT C is more likely to come from diet or supplements??
  • Line 360/361 :In my opinion, the authors should attenuate their statement about the CUP project. This study adds to the evidence of the CUP project… (more powerful should be deleted).

Author Response

Reviewer 1 comments:

This is a well written extensive systematic review and meta-analyses of cohort studies on dietary factors and breast cancer prognosis. In my opinion, it adds to the evidence provided bij the CUP (continuous update project) of the WCRF.

Comment 1: The influence of confounding factors is rarely discussed in the paper. The authors should elaborate on this. How was the confounding adjustment in the cohort studies? Did the studies adjust for example for physical activity? Is confounding incorporated in the quality adjustment? In the discussion section, the authors should discuss whether (residual confounding) could have influenced the results.

Response 1:

The analyses of the cohort studies in this paper were adjusted for several different confounding factors. Most study analyses were adjusted for age, energy intake, and lifestyle factors, such as smoking, physical activity, and alcohol consumption. Moreover, the analyses of some studies that recruited multiethnic or multiresidential participants were adjusted for race/ethnicity and residency. We used the Newcastle-Ottawa Scale (NOS) scoring system to assess the quality of the cohort studies included in the analysis. In the NOS system, there is a comparability section, which includes confounding factors, such as age or gender, as well as additional factors. These confounding factors and NOS quality items are displayed in Tables S1 and S2. Therefore, the analysis of most of the cohort studies was adjusted for several residual confounding factors, including lifestyle factors (smoking, physical activity, and alcohol consumption) that can negatively influence cancer incidence and total energy intake to determine the precise dietary assessment. Given that most residual confounding factors were already controlled for, we did not identify any confounding factors that may have influenced the result errors in this paper.

Comment 2: The authors describe in the introduction (line 59/60) and discussion (line 356/357) sections that RCTs might easily lead to selection bias. I am not convinced of this statement. RCTs might another way to test mechanisms and effects of specific diets or dietary components and for example supplements. A RCT can lead to selection bias by selective loss to follow-up. But is that what the authors mean? Or do they mean RCTs include selective populations with a healthier background diet? In the same sentences, the authors state that case-control studies belong to the experimental study designs. This is not correct and should be adjusted.

Response 2:

Thanks for your useful critique. In line 59/60 and line 356/357, I have edited the text to reduce ambiguity. The original meaning of this sentence was that RCT and case-control designs may lead to selection bias during the process of participant recruitment. For example, participants in the control group in these study designs are generally recruited from a healthy population, and this may generate selection bias. In the field of nutritional epidemiology, recent trends have shown a preference for prospective cohort studies with adequate characteristics rather than case-control studies to demonstrate stronger evidence [1]. Accordingly, I replaced the sentence with “Because most observational studies have selection bias due to participant recruitment methods, nutritional epidemiological analyses generally include only prospective cohort studies.” In line 59/61.

Comment 3: The NOS should include a clear legend; what do all items and scores mean? Why are studies with scores below 6 considered as low-quality studies. What is the evidence for this cut-off?

Response 3:

We displayed each NOS item for the cohort studies in Table S2. NOS items for cohort studies are largely divided into three parts: selection, comparability, and outcome. In the selection section, there are four different items: representativeness of the exposed cohort, selection of the nonexposed cohort, ascertainment of exposure, and outcome of interest not present at the start of the study. In the comparability section, there are two different items: age or gender, and additional factors. Last, in the outcome section, there are three different items: assessment of outcome, follow-up length, and adequacy of follow-up of cohorts. In each quality assessment category, a positive answer is marked with an asterisk (*) and earns 1 point. The NOS for cohort studies has a total of 9 items; therefore, the maximum possible score is nine. We attached the NOS items below.

Additionally, we generally judged whether the studies were qualified considering the criteria of the average score of whole studies. For example, if the average score was 7, we selected studies with scores greater than 7 for meta-analysis. Another method judges study quality as ‘good’, ‘fair’, or ‘poor’. Good quality is considered when a score of 3 or 4 is achieved in the selection domain, 1 or 2 is achieved in the comparability domain, and 2 or 3 is achieved in the outcome domain. Fair quality is considered when a score of 2 is achieved in the selection domain, 1 or 2 is achieved in the comparability domain, and 2 or 3 is achieved in the outcome domain. Last, poor quality is considered when a score of 0 or 1 is achieved in the selection domain, 0 is achieved in the comparability domain, and 0 or 1 is achieved in the outcome domain. In our current study, we followed the latter method to judge study quality. If the study was of good quality, scores would range from 6 to 9. Accordingly, we decided that good-quality studies (6-point studies) should be included in the meta-analysis in this paper. These methods are detailed at http://www.ohri.ca/programs/clinical_epidemiology/oxford.asp.

Comment 4: 2.5. statistical analysis. Is it possible to add dose-response analyses?

Response 4:

             According to the reviewer’s comment, we wanted to conduct dose-response meta-analyses of the major dietary factors that showed significant differences in this study. However, some major dietary factors that can influence breast cancer prognosis, have previously been published by assessment of dose-response meta-analysis (for example, soy isoflavones and soy protein in line 386/390). Moreover, we could not conduct dose-response meta-analysis, because we had not sufficient study numbers to assess dose-response meta-analysis without any detail information of an intake amount (line 437/439 and line 533/536). Given these reasons, we already suggested the need for further dose-response meta-analyses for major dietary factors that can influence breast cancer prognosis as a study limitation in the discussion section (line 531/534). In other words, we will perform a future dose-response meta-analysis for the major dietary factors affecting the prognosis of breast cancer as suggested by the reviewer.

Comment 5: 3.2. study characteristics (line 158/159): 120,000 women in total & 9,659 pre-menopausal and 35,366 postmenopausal women -> is the menopausal status of the remaining women unclear? Please explain.

Response 5:

             The sum of premenopausal and postmenopausal women was not same as the total number of breast cancer survivors. The reason was that most studies did not precisely divide the population into premenopausal or postmenopausal women; these studies assessed a mixed population of premenopausal and postmenopausal women. Therefore, we calculated the total number of breast cancer survivors as the sum of the total population including a mixture of premenopausal and postmenopausal women, and the number of premenopausal or postmenopausal women was calculated for only precisely specified populations of premenopausal and postmenopausal women. Therefore, the sum of premenopausal women and postmenopausal women was less than the sum of the total breast cancer survivors.

Comment 6: 3.2. study characteristics (line 159/160): The total number of deaths is lower than the BC-specific deaths? Please adjust.

Response 6:

             Similar to the response to comment #5, studies that were assessed in our paper were focused on BC-specific deaths rather than total deaths (overall mortality). Therefore, the number of BC-specific deaths was not the same as the number of total deaths or did not include the number of total deaths.

Comment 7: line 270: the intake of rye bread appeared positively associated with all-cause mortality in postmenopausal women, while this is a healthy product with a high fiber load. Can the authors explain this association? Chance finding? Confounding?

Response 7:

             Our results showed that the intake of rye bread was positively associated with all-cause mortality, even though this food is considered a healthy product with a high fiber load. This relationship between rye bread and BC prognosis was assessed in a Danish cohort with different intake times: before and after breast cancer diagnosis. Therefore, the study number did not allow meta-analysis, which was a limitation. Additionally, it is hard to formulate conclusive results. In the Danish cohort study, they referred to rye bread as a representative example of a whole-grain product that is rich in fiber. In our systematic review and meta-analysis, there was no significant correlation between the intake of whole grain products and breast cancer prognosis, while the intake of fiber alleviated the risk of all-cause mortality. Therefore, while the intake of fiber has some preventive effects, the intake of whole-grain products had no effect on the risk of mortality. Taken together, we have determined that the intake of rye bread among Danish people is one of the risk factors for mortality. No biological mechanisms have yet been elucidated to explain why the intake of rye bread increased the risk of breast cancer-associated mortality, and we estimate that most Danish people use rye bread to make open sandwiches, which is easy to consume as a regular meal. An open sandwich on rye bread may contain animal fats from sandwich toppings, which may detrimentally influence breast cancer prognosis. Accordingly, we suggest that rye bread is not considered a contributing food source for breast cancer mortality.

Comment 8: line 303/304: VIT C is more likely to come from diet or supplements?

Response 8:

             According to your comment, in lines 303/304, we described vitamin C intake via both diet and supplements. As described below, we explained in detail that dietary vitamin C intake would have an effect on the decreased risk of BC-specific mortality, while vitamin C supplements would like to prevent the risk of recurrence (line 317/320).

Comment 9: line 360/361: In my opinion, the authors should attenuate their statement about the CUP project. This study adds to the evidence of the CUP project (more powerful should be deleted).

Response 9:

             As your suggestion, we deleted part of the statement about our study’s superiority over the CUP project.

Reviewer 2 Report

Patients always ask if there is any recomendded diet after diagnosis off breast cancer. I think this  information is  very important. A supplement directed to patient that will guide them as to the benefit of som, food will be of great value.

Author Response

Reviewer 2 comments: Patients always ask if there is any recommended diet after diagnosis off breast cancer. I think this information is very important. A supplement directed to patient that will guide them as to the benefit of som, food will be of great value.

Response:

             We appreciate your compliments regarding our study. This study contributes to the identification of effective foods and dietary components to improve prognosis in breast cancer patients. Therefore, we will continue to contribute to providing patients with dietary information that can improve breast cancer prognosis by conducting further studies that are presented in the discussion section.
